# 3D Vision-Language Gaussian Splatting

**Qucheng Peng[1,2]\*, Benjamin Planche[2]†, Zhongpai Gao[2], Meng Zheng[2], Anwesa Choudhuri[2], Terrence Chen[2], Chen Chen[1], Ziyan Wu[2]**

[1]Center for Research in Computer Vision, University of Central Florida, Orlando, FL, USA
[2]United Imaging Intelligence, Boston, MA, USA
qucheng.peng@ucf.edu, {first.last}@uii-ai.com, chen.chen@crcv.ucf.edu

## ABSTRACT

Recent advancements in 3D reconstruction methods and vision-language models have propelled the development of multi-modal 3D scene understanding, which has vital applications in robotics, autonomous driving, and virtual/augmented reality. However, current multi-modal scene understanding approaches have naively embedded semantic representations into 3D reconstruction methods without striking a balance between visual and language modalities, which leads to unsatisfying semantic rasterization of translucent or reflective objects, as well as over-fitting on color modality. To alleviate these limitations, we propose a solution that adequately handles the distinct visual and semantic modalities, i.e., a 3D vision-language Gaussian splatting model for scene understanding, to put emphasis on the representation learning of language modality. We propose a novel cross-modal rasterizer, using modality fusion along with a smoothed semantic indicator for enhancing semantic rasterization. We also employ a camera-view blending technique to improve semantic consistency between existing and synthesized views, thereby effectively mitigating over-fitting. Extensive experiments demonstrate that our method achieves state-of-the-art performance in open-vocabulary semantic segmentation, surpassing existing methods by a significant margin.

## 1 INTRODUCTION

The advancement of 3D reconstruction methods, such as neural radiance fields (NeRF) (Mildenhall et al., 2020) and 3D Gaussian splatting (3DGS) (Kerbl et al., 2023), has enabled the effective acquisition of *3D color representations*, facilitating high-fidelity and real-time rendering from novel viewpoints. Moreover, vision-language models like CLIP (Radford et al., 2021) and LSeg (Li et al., 2022) have been bridging the gap between color images and semantic features in 2D space. Given an input image, these models can generate a dense 2D language map, *i.e.*, assigning semantically-rich language embeddings to each pixel (*e.g.*, a pixel depicting a person's face can be assigned a language embedding de-

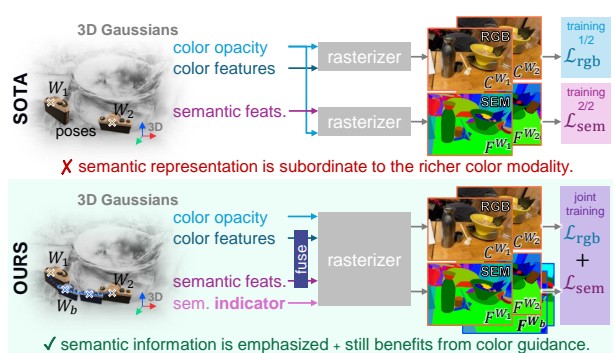

Figure 1: Comparison of prior semantic 3DGS work and our novel method. We apply cross-modal rasterization and camera-view-based regularization for better exploration of semantic features.

scribing said face). Building on these developments, multi-modal 3D scene understanding—which aims to learn effective *3D semantic representations* from multi-view images and their corresponding camera poses—has made significant progress in recent years. This area of research has a wide range of applications across various practical domains, including robotic manipulation (Shorinwa et al., 2024), autonomous driving (Gu et al., 2024), and VR/AR (Guerroudji et al., 2024).

---

*This work was done during Qucheng Peng's internship at United Imaging Intelligence, Boston, MA, USA.
†Corresponding author.

Recent methods (Engelmann et al., 2024; Qin et al., 2024; Shi et al., 2024; Zhou et al., 2024b; Yu et al., 2024a) in multi-modal 3D scene understanding have adopted the paradigm of *embedding* semantic representations into 3D representation for joint reconstruction training, and utilizing semantic knowledge distilled from off-the-shelf vision-language models to guide the training process. For example, OpenNeRF (Engelmann et al., 2024) integrates MipNeRF (Barron et al., 2021) method with OpenSeg model (Ghiasi et al., 2022), and LangSplat (Qin et al., 2024) employs both 3DGS (Kerbl et al., 2023) method and CLIP (Radford et al., 2021) model. These solutions rely on 2D supervision to learn a multi-modal (color and semantic) 3D scene representation, *i.e.*, projecting the learned 3D representation back to 2D views for comparison with the input observations (input color images and corresponding 2D language maps inferred by aforementioned vision-language models).

However, we argue that these methods have *naively embedded* semantic representations into 3D reconstruction methods like 3DGS—originally designed for color information—failing to strike a balance between visual and language modalities. For example, they are directly applying the color rasterization function—meant to project 3D RGB information to 2D—to the new language modality, ignoring that this function relies on a color opacity attribute that does not translate to semantic information. Prior art also tends to ignore the unequal complexity and distribution of color and semantic modalities, and the risk of over-fitting the color information to the detriment of the 3D semantic representations. While the same objects may exhibit different colors from various views, their semantic information remains constant. Conversely, different objects can share similar colors, but it is less desirable for their semantic representations to appear identical. Thus, training color representations may negatively impact the training of 3D semantic representations.

Given these limitations, our intuition is to strike a *balance* between visual and language modalities, rather than simply embedding language features into RGB-based 3D reconstruction. Therefore, we propose a novel framework named 3D vision-language Gaussian splatting, as shown in Fig. 1. On one hand, we propose a novel cross-modal rasterizer that prioritizes the rendering of language features. We integrate semantic features with meaningful information from the color domain through modality fusion, prior to rasterization, to facilitate the robust learning of semantic information. Besides, we introduce a language-specific parameter that enables the meaningful blending of language features from different Gaussians. This methodology yields a more accurate representation of semantic information, especially for translucent or reflective objects, such as glass and stainless steel, where the usage of color opacity may lead to misinterpretation. On the other hand, we also propose a novel camera-view blending augmentation scheme specific to the semantic modality, *i.e.*, blending information across views to synthesize new training samples. This process regularizes the language modality through enhancing semantic consistency between the existing and novel views, leading to more robust 3D semantic representations.

All in all, our 3D vision-language Gaussian splatting can be summarized into the following contributions:

- We propose a cross-modal rasterizer that places greater emphasis on language features. Modality fusion occurs prior to rasterization, accompanied by a learnable and independent *semantic indicator* parameter for the $\alpha$-blending of language features, enabling a more accurate representation of translucent or reflective objects.
- We define a camera-view blending technique for the regularization of semantic representations during training, augmenting the input 2D language maps through a cross-modal view, *i.e.*, leveraging the semantic-consistency prior to alleviate over-fitting on the color modality.
- Extensive experiments on benchmark datasets demonstrate that our approach achieves state-of-the-art performance in open-vocabulary semantic segmentation tasks, outperforming existing methods by a significant margin.

## 2 RELATED WORK

**3D Neural Representations.** Recent advancements in 3D scene representation have made significant progress, particularly with neural radiance fields (NeRF) (Mildenhall et al., 2020), which excel in novel view synthesis. However, NeRF's reliance on a neural network for implicit scene representation can result in prolonged training and rendering times. Methods like Instant-NGP (Müller et al., 2022) speed up these processes through hash encoding, while approaches such as 3D Gaussian splatting (3DGS) (Kerbl et al., 2023) use explicit neural representations for better alignment with

GPU computations via differential tile rasterization. Additionally, MipNeRF (Barron et al., 2021) and Mip-Splatting (Yu et al., 2024b) address aliasing issues, whereas DS-NeRF (Deng et al., 2022) and DG-Splatting (Chung et al., 2024) focus on sparse view reconstructions. *In this paper, we utilize 3D Gaussian Splatting for 3D neural representations.*

**Visual Foundation Models.** Visual foundation models include both pure vision models, crucial for semantic segmentation and feature extraction, and vision-language models that connect images with natural language. Among pure vision models, the Segment Anything Model (SAM) (Kirillov et al., 2023) excels in zero-shot transfer, generating multi-scale segmentation masks. DINO (Caron et al., 2021) and DINOv2 (Oquab et al., 2024), trained in a self-supervised manner, deliver fine-grained features for various downstream tasks. In vision-language models, CLIP (Radford et al., 2021) utilizes contrastive learning to align visual and textual features, while LSeg (Li et al., 2022) enhances this by incorporating spatial regularization for refined predictions. APE (Shen et al., 2024) functions as a universal perception model for diverse multimodal tasks. *In this paper, we employ SAM and CLIP to extract ground-truth features for baseline comparisons, ensuring a fair evaluation.*

**Multi-modal 3D Scene Understanding.** Existing methods for scene understanding learn multi-modal 3D representations from posed images, enabling rendering from novel viewpoints for tasks like open-vocabulary semantic segmentation. For NeRF-based approaches, LERF (Kerr et al., 2023) optimizes a language field alongside NeRF, using positions and physical scales to generate CLIP vectors. OpenNeRF (Engelmann et al., 2024) introduces a mechanism for obtaining novel camera poses, enhancing feature extraction. Among 3DGS-based methods, LangSplat (Qin et al., 2024) adopts a two-stage strategy for semantic feature acquisition, while GS-Grouping (Ye et al., 2024) extends Gaussian splatting for joint reconstruction and segmentation. Additionally, GOI (Qu et al., 2024) employs hyperplane division to select features for improved alignment. Moreover, HUGS Zhou et al. (2024a) enables holistic 3D scene understanding by integrating 2D semantics and flow with 3D tracking, effectively lifting them into the 3D space.

## 3 METHODOLOGY

### 3.1 PROBLEM STATEMENT

According to the vanilla Gaussian splatting (3DGS) paradigm applied to RGB image rendering (Kerbl et al., 2023), a scene is represented by a set of 3D Gaussians $\mathcal{G} = \{g^i\}_{i=1}^{\mathcal{N}}$, where $\mathcal{N}$ denotes their number. Each Gaussian is defined as $g^i = \{\mu^i, \Sigma^i, o^i, c^i\}$, *i.e.*, by its mean position $\mu^i \in \mathbb{R}^3$, covariance matrix $\Sigma^i \in \mathbb{R}^{3 \times 3}$, opacity $o^i \in \mathbb{R}$, and color properties $c^i \in \mathbb{R}^{d_c}$ (*e.g.*, with $d_c = 3$ for RGB parameterization). Images are rasterized by splatting Gaussians through each pixel $v$ into the scene and $\alpha$-blending the Gaussian contributions to the color $C(v)$, as:

$$C(v) = \sum_{i=1}^{\mathcal{N}} c^i o^i P^i \prod_{j=1}^{i-1}(1 - o^j P^j) \;\; \text{with} \;\; P^i = e^{-\frac{1}{2}(v-\widehat{\mu}^i)(\widehat{\Sigma}^i)^{-1}(v-\widehat{\mu}^i)}, \tag{1}$$

where $\widehat{\mu}^i$ and $\widehat{\Sigma}^i$ are the 3D-to-2D projections of $\mu^i$ and $\Sigma^i$. This scene representation is learned from a training set $T^r = \{(I_1^r, W_1^r), (I_2^r, W_2^r), \ldots\}$ consisting of several pairs of RGB images $I$ of the target scene and the corresponding camera poses $W$.

Gaussian splatting models have been expanded to incorporate language-embedding information densely describing the scene (Qin et al., 2024; Zhou et al., 2024b; Qu et al., 2024) by adding a language feature vector $f^i \in \mathbb{R}^{d_h}$ (with $d_h$ feature size) to each 3D Gaussian and similarly rasterizing this modality. *I.e.*, a 2D semantic embedding $F$ at pixel $v$ can be expressed as:

$$F(v) = \sum_{i=1}^{\mathcal{N}} f^i o^i P^i \prod_{j=1}^{i-1}(1 - o^j P^j). \tag{2}$$

Even though the two modalities have widely different properties (*e.g.*, translucence only applies to the visual modality and not the semantic one), previous methods have been directly applying the color rasterization process to the language features without any adaption, *i.e.*, only replacing $c^i$ (Eq. 1) by $f^i$ (Eq. 2). In this paper, we propose to adapt the usual rasterization scheme to better fit the language-feature modality.

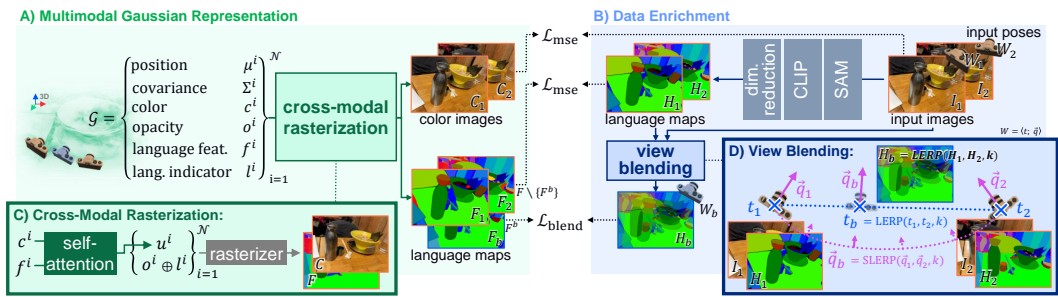

Figure 2: Overview of our proposed framework. A) We propose a novel multi-modal Gaussian splatting model; B) we enrich the input images and poses for the model to better fit the semantic information. Besides our introduction of a novel semantic indicator parameter $l$, our additional contributions are: C) a semantic-aware cross-modal rasterization module; and D) a camera view blending augmentation scheme for training regularization.

To train these semantically-enriched 3DGS models, the standard procedure consists of first generating the set of 2D language-feature maps $H$ corresponding to the input images $I$. Off-the-shelf pure vision models like SAM (Kirillov et al., 2023) and vision-language models like CLIP (Radford et al., 2021) are typically used, along with an auto-encoder-based Zhai et al. (2018) dimension reduction as in Qin et al. (2024). Once the data is prepared, the 3D Gaussians can be iteratively optimized by minimizing the distance between its rasterized 2D semantic embeddings ($c.f.$ Eq. 2) and the ground-truth 2D semantic embeddings:

$$\mathcal{L} = \mathbb{E}_{(I,W) \in T^r} \mathbb{E}_{v \in I} \mathcal{L}_{\text{sem}}(F^W(v), H^W(v)), \tag{3}$$

where $\mathcal{L}$ is the overall optimization objective. Besides, $F^W$ and $H^W$ are the predicted and ground-truth 2D language maps rendered using camera pose $W$ corresponding to image $I$ (for ease of readability, we drop the superscript $W$ in subsequent equations), and $\mathcal{L}_{\text{sem}}$ is a distance function for 2D semantic maps ($e.g.$, L1 distance).

The resulting 3D Gaussian language representation can be leveraged for open-vocabulary semantic queries. For example, given a query language vector $\tau$, a pixel-wise 2D relevancy score map corresponding to view $W$ can be computed as (Radford et al., 2021):

$$p(\tau|v) = \exp(\frac{F(v) \cdot \varphi(\tau)}{\|F(v)\|\|\varphi(\tau)\|}) / \sum_{s \in \mathcal{T}} \exp(\frac{F(v) \cdot \varphi(s)}{\|F(v)\|\|\varphi(s)\|}), \tag{4}$$

where $\varphi$ is the text encoder from the vision-language model. Such a relevancy map can be leveraged, $e.g.$, for open-vocabulary 2D localization ($i.e.$, by measuring the argmax response to the query) or semantic segmentation ($i.e.$, by thresholding the resulting relevancy map).

## 3.2 SEMANTIC-AWARE RASTERIZATION

Existing multi-modal rasterization methods (Qin et al., 2024; Zhou et al., 2024b; Zuo et al., 2024) have largely drawn from color-based 3DGS (Kerbl et al., 2023). These rasterizers approach the rendering of 3D semantic information in a manner akin to RGB rendering (see Sec. 3.1), resulting in an insufficient focus on semantic-specific design. To address this gap, we propose a novel cross-modal rasterizer that emphasizes semantic-specific design, as illustrated in Fig. 2A and Fig. 2C.

A first noticeable shortcoming is the insufficient integration and exchange between 3D semantic and color features. While these modalities have distinct properties, they offer correlated and complementary information about the scene, with knowledge from one informing the other. Current models fail to leverage this synergy, resulting in inadequate guidance for learning semantic representations from the richer color modality.

To address this shortcoming, we propose integrating 3D semantic and color features prior to rasterization, rather than treating them independently (Qin et al., 2024; Qu et al., 2024), which facilitates effective knowledge fusion and exchange between modalities. This intuitive improvement to multi-modal 3DGS has surprisingly been ignored in prior work. We decide to correct this oversight

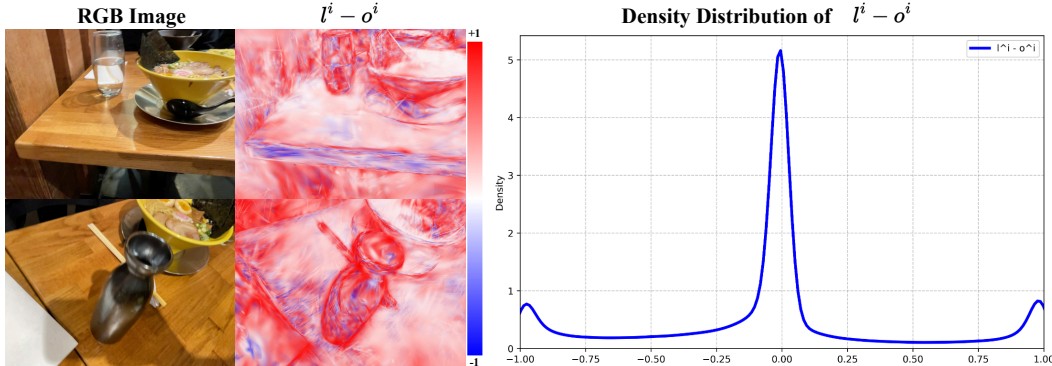

Figure 3: Empirical differences between color opacity and proposed smoothed semantic indicator. On the left, we visualize the difference $l^i - o^i$ in Gaussians modeling the `ramen` scene. While their color opacity may vary significantly, most Gaussians need their semantic features to be rasterized with minimal blending (*c.f.* red regions in the difference maps, *i.e.*, where $l^i \gg o^i$), except for Gaussians representing intangible lighting effects (*c.f.* glares on the bottle, bowl, table, *etc*.). On the right, we further plot the density distribution of $l^i - o^i$ in Gaussians for the `ramen` scene, indicating the different distributions of these two control parameters.

by applying the well-established self-attention mechanism for modality fusion before rasterization, thereby enhancing its effectiveness. We can derive the 3D modality-fused features $u^i \in \mathbb{R}^{d_c+d_f}$ for our multi-modal rasterizer as:

$$u^i = c^i \oplus f^i + \psi_{\text{out}}(\text{softmax}(\frac{\psi_Q(c^i \oplus f^i)\psi_K^\top(c^i \oplus f^i)}{\sqrt{d_h}})\psi_V(c^i \oplus f^i)), \quad (5)$$

where $d_h$ represents the number of heads. Moreover, $\psi_Q, \psi_K, \psi_V : \mathbb{R}^{d_c+d_f} \mapsto \mathbb{R}^{d_c+d_f} \times \mathbb{R}^{d_h}$ and $\psi_{\text{out}} : \mathbb{R}^{d_c+d_f} \times \mathbb{R}^{d_h} \mapsto \mathbb{R}^{d_c+d_f}$ are single-layer linear networks.

Another salient shortcoming is the simplistic adoption of $\alpha$-blending-based RGB splatting (Eq. 1) to rasterize language embeddings, only substituting the 3D color features $c$ with 3D semantic representations $f$ (Eq. 2). In other words, previous works use the opacity attribute of Gaussians to render not only RGB images but also language maps, *c.f.* Eq. 2 (either optimizing the opacities values over the two modalities (Shi et al., 2024; Zhou et al., 2024b), or applying frozen opacity values from the RGB-only pre-training during semantic rasterization (Qin et al., 2024)). We argue that this approach fails to recognize that in phenomena involving semi-opaque media (*e.g.*, glass, water) and complex light transport effects (*e.g.*, direction-dependent scattering and glare), color opacity cannot be effectively translated to the semantic modality. In most cases, when Gaussians represent *tangible* elements of the scene (*e.g.*, the surface or volume of an object), their semantic features should be splatted to 2D without any reduction in opacity. Conversely, *intangible* Gaussians (*e.g.*, those simulating lighting effects like optical glares) should be excluded from semantic rasterization by enforcing their $\alpha$-blending weight close to zero.

To address this problem, we propose a novel $\alpha$-blending strategy specifically designed for exploring semantic information. We introduce a new learnable attribute for our multi-modal Gaussians: a **smoothed semantic indicator** $l^i \in [0, 1]$, which is applied to the rasterization of language embeddings, effectively replacing the color opacity parameter for this modality:

$$U(v) = \underbrace{\sum_{i=1}^{\mathcal{N}} u_{1:d_c}^i o^i P^i \prod_{j=1}^{i-1}(1 - o^j P^j)}_{\text{color modality (channels 1 to } d_c\text{)}} \oplus \underbrace{\sum_{i=1}^{\mathcal{N}} u_{d_c+1:d_c+d_f}^i l^i P^i \prod_{j=1}^{i-1}(1 - l^j P^j)}_{\text{language modality (channels } d_c+1 \text{ to } d_c+d_f\text{)}}. \quad (6)$$

*Note that these two processes are conducted simultaneously in the rasterizer.* By disentangling the semantic rasterization from the opacity-based control and letting the overall model learns an independent per-Gaussian semantic indicator, we allow the model to better fit the semantic information of the scene. This is highlighted in Fig. 3, which shows how much the distribution of semantic indicator values differs from the color opacity after optimization. This is especially notable for

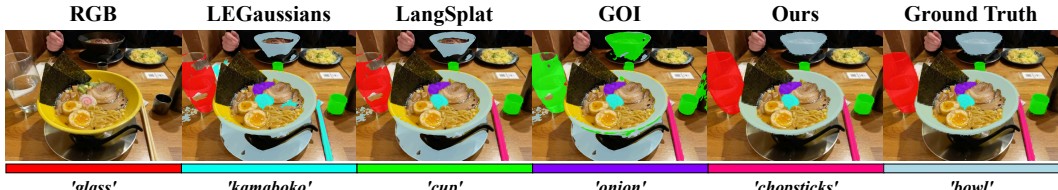

Figure 4: Qualitative semantic segmentation comparisons on the `ramen` scene (LERF dataset).

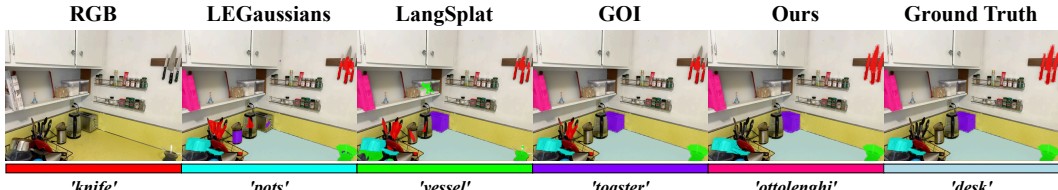

Figure 5: Qualitative semantic segmentation comparisons on the `kitchen` scene (LERF dataset).

semi-transparent or reflective objects (*e.g.*, glass and pot in depicted scene), exhibiting low opacity but high semantic-indicator values (*c.f.* $l^i - o^i$ close to 1).

After rasterization, the resulting pixel information $U(v)$ can be further decomposed as:

$$U(v) = C(v) \oplus F(v), \tag{7}$$

where $C(v)$ and $F(v)$ are the RGB color and 2D language embeddings at pixel $v$ respectively.

Based on the above, we formulate the loss function to optimize the overall scene representation as follows, $\forall (I, W) \in T^r$:

$$\mathcal{L}_{\text{raster}}(W) = \mathbb{E}_{v \in I} \left[ \mathcal{L}_{\text{MSE}} \left( C^W(v), I(v) \right) + \mathcal{L}_{\text{MSE}} \left( F^W(v), H^W(v) \right) \right], \tag{8}$$

where $\mathcal{L}_{\text{MSE}}$ represents the mean squared error (MSE) criterion.

### 3.3 SEMANTIC-AWARE CAMERA VIEW BLENDING

While the novel cross-modal rasterization process proposed in Sec. 3.2 facilitates the efficient learning of joint color and semantic representations, this multi-modal approach may be suboptimal for applications focused exclusively on semantic information—such as open-vocabulary semantic segmentation tasks—due to the adverse effect from richer color modality. On the one hand, though the same object may exhibit different colors from various viewpoints, its semantic information remains consistent. On the other hand, different objects might share similar colors, which can lead to undesirable identical semantic representations. Therefore, it is crucial to incorporate regularization methods that enhance semantic consistency across diverse views, thereby mitigating the overfitting on semantic representations.

To this end, we propose a regularization technique called camera view blending (shown in Fig. 2B), designed specifically to augment the semantic modality by leveraging multi-modal information. Inspired by mix-up augmentation Zhang et al. (2018), our approach involves blending two ground-truth 2D language maps to create new ones, while providing semi-consistent corresponding camera poses through advanced 6DOF pose interpolation. For two randomly selected samples $(I_1, W_1)$ and $(I_2, W_2)$ from the training set $T^r$, where $W_1 \neq W_2$, we first utilize the camera poses $W_1$ and $W_2$ to synthesize a novel camera pose that incorporates both rotation and translation components.

Expressing the input rotations with quaternions $q_1$ and $q_2$, we apply spherical linear interpolation (Slerp) (Shoemake, 1985). Define $\theta = \arccos \frac{q_1 \cdot q_2}{\|q_1\| \|q_2\|}$ and let the interpolation ratio $k$ be randomly sampled from Beta$(0.2, 0.2)$. The quaternion for the novel camera view $q_b$ is then given by:

$$q_b = \begin{cases} k \dfrac{q_1}{\|q_1\|} \cdot \dfrac{\cos\theta}{|\cos\theta|} + (1-k)\dfrac{q_2}{\|q_2\|} & \text{if } |\cos\theta| > 0.995 \\[2ex] \dfrac{q_1 \sin(\theta - k\theta)}{\|q_1\| \sin\theta} \cdot \dfrac{\cos\theta}{|\cos\theta|} + \dfrac{q_2 \sin(k\theta)}{\|q_2\| \sin\theta} & \text{if } |\cos\theta| \leq 0.995. \end{cases} \tag{9}$$

When the two quaternions are nearly aligned in direction ($|\cos\theta| > 0.995$), we use linear interpolation (Lerp) for improved efficiency. If they are not aligned ($|\cos\theta| \leq 0.995$), Slerp is applied.

For the translation component, we use linear interpolation to synthesize the focal center of the novel view. Given the camera centers $t_1$ and $t_2$ in the world coordinate system, corresponding to $W_1$ and $W_2$ respectively, the new center is determined with the same interpolation ratio $k$:

$$t_b = kt_1 + (1-k)t_2. \tag{10}$$

With the blended quaternion $q_b$ and camera center $t_b$, we empirically derive the novel camera view $W_b$, which can be used to render the associated 2D semantic embeddings from the Gaussian model. Our goal is to enhance the semantic consistency across existing views and their associated synthesized views, thereby regularizing the training of 3D semantic representations. Thus, the camera view blending loss $\mathcal{L}_b$ for semantics with identical interpolation ratio $k$ is:

$$\mathcal{L}_b = \text{SSIM}(I_1, I_2)\mathbb{E}_{v \in I_1}\mathcal{L}_{\text{MSE}}(F^{W_b}(v), H^{W_b}(v)) \text{ with } H^{W_b} = kH^{W_1} + (1-k)H^{W_2}. \tag{11}$$

Here $F^{W_b}(v)$ denotes the rendered 2D semantic embeddings based on $W_b$ using the rasterizer described in Section 3.2, while $H^{W_1}(v)$ and $H^{W_2}(v)$ represent the ground truth 2D semantic features for $W_1$ and $W_2$ respectively. When the two images differ significantly, regularization may be counterproductive. To address this, we utilize the Structural Similarity Index Measure (SSIM) between $I_1$ and $I_2$ as a weighting factor. By enforcing similarity between $H^{W_b}(v)$ and $F^{W_b}(v)$, we enhance the semantic consistency of the 3D representations, thereby mitigating the overfitting issue.

By combining the rasterization loss (Eq. 8) and the camera view blending loss (Eq. 11) according to a trade-off hyper-parameter $\lambda$, the overall objective is represented as:

$$\mathcal{L}_{\text{overall}} = \mathbb{E}_{(I_1, W_1),(I_2, W_2) \in T^r}[\mathcal{L}_{\text{raster}}(W_1) + \mathcal{L}_{\text{raster}}(W_2) + \lambda\mathcal{L}_b(W_1, W_2)]. \tag{12}$$

## 4 EXPERIMENTS

### 4.1 SETTINGS

**Datasets.** We employ 3 datasets for our evaluation on open-vocabulary semantic tasks. (1) **LERF** dataset (Kerr et al., 2023), captured using the Polycam application on an iPhone, comprises complex, in-the-wild scenes and is specifically tailored for 3D object localization tasks. We report the mean Intersection over Union (mIoU) results, alongside localization accuracy results in accordance with (Qin et al., 2024) (2) **3D-OVS** dataset (Liu et al., 2023) consists of a diverse collection of long-tail objects in various poses and backgrounds. This dataset is specifically developed for open-vocabulary 3D semantic segmentation and includes a complete list of categories. Following (Qin et al., 2024; Zuo et al., 2024), the mIoU metric is applied for this dataset. (3) **Mip-NeRF 360** dataset (Barron et al., 2022) contains 9 scenes, each composed of a complex central object or area and a detailed background. Following Qu et al. (2024), we provide some evaluations on this dataset in annex.

**Implementation.** To extract ground-truth semantic features from each image, we utilize the SAM ViT-H model (Kirillov et al., 2023) alongside the OpenCLIP ViT-B/16 model (Radford et al., 2021). The 3D Gaussians for each scene are initialized using sparse point clouds derived from Structure from Motion. For modality fusion, we set $d_c$ and $d_f$ to 3, while $d_h$ is set to 4. During rasterization, smoothed semantic indicator is initialized in the same manner as color opacity. For each iteration, two camera views and their associated images are selected, and 3D Gaussians are trained for 15,000 iterations. Moreover, the parameter $\lambda$ in Equation 12 is configured to 1.2. The implementation of semantic opacity is done in CUDA and C++, while the other components are in PyTorch. All experiments are conducted on Nvidia A100 GPUs. For each scene, our model is trained for 15,000 iterations using an Adam optimizer (Kingma, 2014), and the learning rates of different components are shown in the appendix. After convergence, the model is applied to the localization and segmentation tasks according to the procedure described in Sec. 3.1.

### 4.2 RESULTS

**Baselines.** For a fair comparison, we select the latest works on open-vocabulary 3D scene understanding: **Feature-3DGS** (Zhou et al., 2024b), **LEGaussians** (Shi et al., 2024), **LangSplat** (Qin et al., 2024), **GS-Grouping** (Ye et al., 2024), **GOI** (Qu et al., 2024), and **FMGS** (Zuo et al., 2024),

all of which are based on 3DGS (Kerbl et al., 2023), and employ the same segmentation and vision-language models mentioned in Sec. 4.1 to extract ground truth language features. *It is important to note that FMGS (Zuo et al., 2024) does not report mIoU results on the LERF dataset and is also not open-sourced, so its mIoU results are not listed.*

**Quantitative Evaluation.** Tab. 1 and Tab. 2 present the mIoU and localization accuracy results on the LERF dataset, respectively. Tab. 3 displays the mIoU results on the 3D-OVS dataset. Our proposed method achieves state-of-the-art performance across all scenes, notably outperforming the second best method LangSplat (Qin et al., 2024) by 10.6 in mIoU on the LERF dataset. Further experiments on other datasets are listed in the appendix.

**Qualitative Evaluation.** To facilitate qualitative comparisons, we contrast our method with several baseline approaches, including LEGaussian (Shi et al., 2024), LangSplat (Qin et al., 2024), and GOI (Qu et al., 2024). Fig. 4 and Fig. 5 illustrate semantic segmentation results on LERF data, while Fig. 6 presents the localization results on the same dataset. For the localization task, the black bounding boxes with dotted lines represent the ground truth ranges, while the red dots indicate predictions from various methods. Besides, segmentation comparisons on the 3D-OVS dataset are displayed in Fig. 7 and Fig. 8. It is clear that our proposed method outperforms the other approaches and is closer to the ground-truth, particularly in reflective and translucent areas. Additional results are displayed in the appendix.

### 4.3 ABLATION STUDIES

**Ablation on cross-modal rasterizer.** Tab. 4.A presents an ablation study on the data-fusion of the rasterizer. In addition to no fusion like LEGaussians (Shi et al., 2024), we also incorporate single-layer MLP and cross-attention for modality Fusion to enhance semantic features for comparison. The results indicate that self-attention modality fusion is the most effective method. Besides, single-layer MLP and cross-attention modality fusion do not demonstrate any advantage over the no fusion condition. In Tab. 4.B, we highlight the benefits of our proposed smoothed semantic indicator. We compare it to previous methods for rasterizing semantic embeddings, specifically by applying the color opacity $o^i$ to the language modality (Qin et al., 2024; Zhou et al., 2024b; Zuo et al., 2024). The results indicate that using a separate attribute for rasterizing language embeddings is clearly advantageous. However, this new attribute cannot be naively fixed, *e.g.*, to 1 or 0.5 for all Gaussians. Its values should be learned by the model through 2D supervision, which is why we introduce a learnable indicator for the $\alpha$-blending of language modality.

**Ablation on camera-view blending.** We also present an ablation of different camera-view blending strategies in Tab. 5, which includes three variants: Rotation that associates to the operation in Eq. 9, Translation that corresponds to the operation in Eq. 10 , and SSIM that weighs Eq. 11. Our proposed method corresponds to the last row of the table. The results indicate that both Slerp-based rotation blending and Lerp-based translation blending contribute to performance improvements. Moreover, the regularization provided by SSIM is essential for controlling the extent of blending.

In Tab. 4.C, we present an ablation study on the interpolation ratio $k$. In addition to fixed values of $0.5$, $0.75$, and $1.0$, we also include experiments based on Even Distribution (U$(0, 1)$), Gaussian Distribution (N$(0, 1)$), and our Beta Distribution (Beta$(0.2, 0.2)$). The results demonstrate that a balanced ratio is crucial for acquiring better representations, and that slight disturbances can be beneficial as well.

Table 1: mIoU results on the LERF dataset.

| Method | Venue | ramen | teatime | figurines | kitchen | avg. |
|---|---|---|---|---|---|---|
| Feature-3DGS | CVPR'24 | 43.7 | 58.8 | 40.5 | 39.6 | 45.7 |
| LEGaussians | CVPR'24 | 46.0 | 60.3 | 40.8 | 39.4 | 46.9 |
| LangSplat | CVPR'24 | 51.2 | 65.1 | 44.7 | 44.5 | 51.4 |
| GS-Grouping | ECCV'24 | 45.5 | 60.9 | 40.0 | 38.7 | 46.3 |
| GOI | ACMMM'24 | 52.6 | 63.7 | 44.5 | 41.4 | 50.6 |
| Ours | | **61.4** | **73.5** | **58.1** | **54.8** | **62.0** |

Table 2: Localization accuracy (%) on the LERF dataset.

| Method | Venue | ramen | teatime | figurines | kitchen | avg. |
|---|---|---|---|---|---|---|
| Feature-3DGS | CVPR'24 | 69.8 | 77.2 | 73.4 | 87.6 | 77.0 |
| LEGaussians | CVPR'24 | 67.5 | 75.6 | 75.2 | 90.3 | 77.2 |
| LangSplat | CVPR'24 | 73.2 | 88.1 | 80.4 | 95.5 | 84.3 |
| GS-Grouping | ECCV'24 | 68.6 | 75.0 | 74.3 | 88.2 | 76.5 |
| GOI | ACMMM'24 | 75.5 | 88.6 | 82.9 | 90.4 | 84.4 |
| FMGS | IJCV'24 | 90.0 | 89.7 | 93.8 | 92.6 | 91.5 |
| Ours | | **92.5** | **95.8** | **97.1** | **98.6** | **96.0** |

Table 3: mIoU results on the 3D-OVS dataset.

| Method | Venue | bed | bench | room | sofa | lawn | avg. |
|---|---|---|---|---|---|---|---|
| Feature-3DGS | CVPR'24 | 83.5 | 90.7 | 84.7 | 86.9 | 93.4 | 87.8 |
| LEGaussians | CVPR'24 | 84.9 | 91.1 | 86.0 | 87.8 | 92.5 | 88.5 |
| LangSplat | CVPR'24 | 92.5 | 94.2 | 94.1 | 90.0 | 96.1 | 93.4 |
| GS-Grouping | ECCV'24 | 83.0 | 91.5 | 85.9 | 87.3 | 90.6 | 87.7 |
| GOI | ACMMM'24 | 89.4 | 92.8 | 91.3 | 85.6 | 94.1 | 90.6 |
| FMGS | IJCV'24 | 80.6 | 84.5 | 87.9 | 90.8 | 92.6 | 87.3 |
| Ours | | **96.8** | **97.3** | **97.7** | **95.5** | **97.9** | **97.1** |

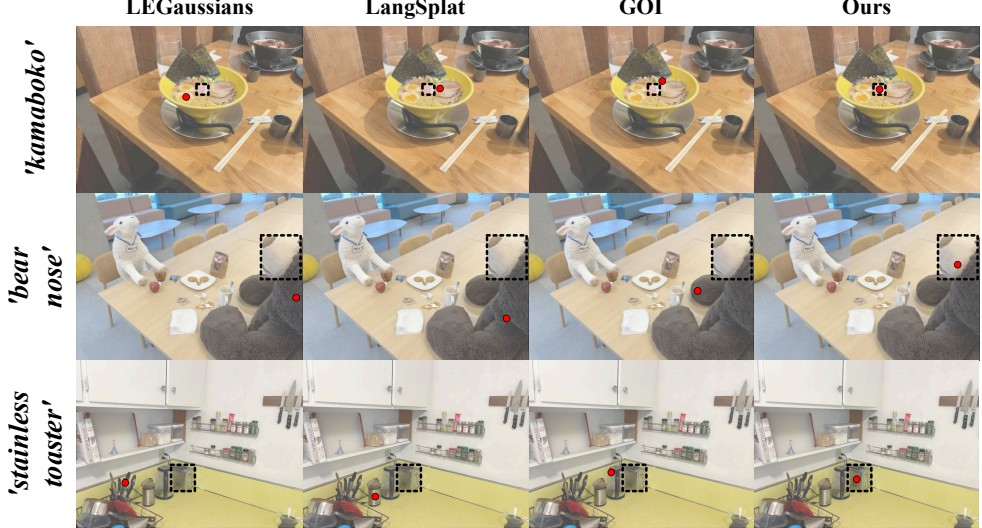

Figure 6: Localization comparisons on LERF. Black dotted-line bounding boxes represent the ground-truth targets, and red dots indicate the predictions (considered correct if within a GT box).

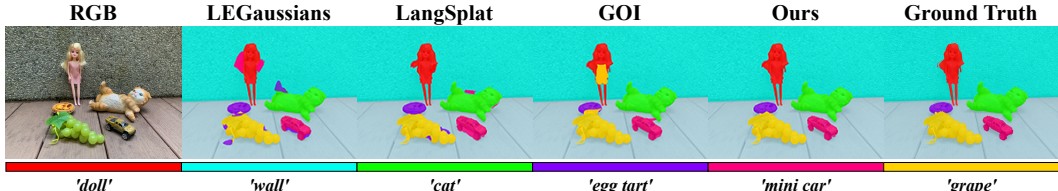

Figure 7: Qualitative semantic segmentation comparisons on the bench scene (3D-OVS dataset).

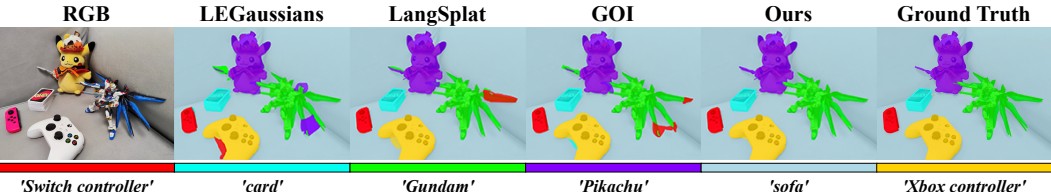

Figure 8: Qualitative semantic segmentation comparisons on the sofa scene (3D-OVS dataset).

Table 4: Ablation results of various contributions on the LERF dataset, in terms of mIoU.

| | Method | ramen | teatime | figurines | kitchen | avg. |
|---|---|---|---|---|---|---|
| (A) Fusion module | No Fusion | 57.8 | 69.6 | 54.2 | 51.5 | 58.3 |
| | Single-layer MLP Fusion | 55.9 | 68.2 | 53.0 | 49.9 | 56.8 |
| | Cross-attention Modality Fusion | 57.3 | 70.7 | 54.6 | 52.0 | 58.7 |
| | Self-attention Modality Fusion (Ours) | **61.4** | **73.5** | **58.1** | **54.8** | **62.0** |
| (B) Indicator value | Fixed to 0.5 | 52.2 | 70.9 | 50.6 | 50.8 | 56.1 |
| | Fixed to 1.0 | 50.3 | 67.7 | 47.1 | 48.5 | 53.4 |
| | Using Color Opacity | 55.9 | 71.0 | 54.2 | 51.3 | 58.1 |
| | Smoothed Semantic Indicator (Ours) | **61.4** | **73.5** | **58.1** | **54.8** | **62.0** |
| (C) Interpolation ratio sampling | Fixed to 0.5 | 58.8 | 72.6 | 56.4 | 52.5 | 60.1 |
| | Fixed to 0.75 | 56.3 | 71.0 | 50.5 | 51.7 | 57.4 |
| | Fixed to 1.0 | 55.2 | 68.3 | 48.5 | 51.0 | 55.8 |
| | Even Distribution | 58.3 | 71.1 | 55.4 | 51.2 | 59.0 |
| | Gaussian Distribution | 57.5 | 71.3 | 53.9 | 50.7 | 58.4 |
| | Beta Distribution (Ours) | **61.4** | **73.5** | **58.1** | **54.8** | **62.0** |

Table 5: Ablation results of Camera View Blending on the LERF dataset, in terms of mIoU.

| Rotation | Translation | SSIM | ramen | teatime | figurines | kitchen | avg. |
|---|---|---|---|---|---|---|---|
| × | × | × | 55.4 | 68.2 | 48.8 | 50.9 | 55.8 |
| × | ✓ | × | 56.6 | 69.5 | 49.7 | 51.5 | 56.8 |
| × | ✓ | ✓ | 58.3 | 70.9 | 51.1 | 52.4 | 58.2 |
| ✓ | × | × | 55.7 | 69.3 | 49.6 | 51.7 | 56.6 |
| ✓ | × | ✓ | 57.9 | 70.2 | 52.3 | 52.1 | 58.1 |
| ✓ | ✓ | × | 59.6 | 71.8 | 53.4 | 53.2 | 59.5 |
| ✓ | ✓ | ✓ | **61.4** | **73.5** | **58.1** | **54.8** | **62.0** |

## 4.4 EFFICIENCY ANALYSIS

Finally, we demonstrate that, despite the new parameter $l$ added to the Gaussians, our contributions actually benefit the efficiency of the resulting scene representation, *i.e.*, in terms of convergence speed and rendering time. As shown in Tab. 6, our model outperforms the other four baselines in efficiency. We attribute this performance to our cooperative training over augmented view/pose batches, and to the increased ability of the Gaussians to accurately fit both target modalities.

Table 6: Efficiency analysis on LERF's ramen.

| Method | Training Time ↓ | FPS ↑ | # of Gaussians ↓ |
|---|---|---|---|
| LangSplat | 96min | 40 | 86k |
| GS-Grouping | 130min | 76 | 107k |
| GOI | 73min | 42 | 92k |
| Ours | **65min** | **79** | **80k** |

## 5 CONCLUSION

In this work, we propose 3D visual-language Gaussian splatting for semantic scene understanding, addressing the neglect of language information in current 3DGS approaches. We propose a novel cross-modal rasterizer that performs modality fusion followed by language-specific rasterization, utilizing a smoothed semantic indicator to disable irrelevant Gaussians, *e.g.*, for scenes with complex light transport (reflections, translucence, *etc.*). Moreover, our camera-view blending technique effectively mitigates over-fitting, ensuring semantic consistency across both existing and synthesized viewpoints. Comprehensive experiments validate the effectiveness of our framework, demonstrating significant improvements in open-vocabulary semantic segmentation compared to prior art. Future research will explore further modalities that could enrich the scene representation (*e.g.*, semantic features from mixtures of foundation models), as well as extending this work to dynamic scenes.

**Ethical Statement**

We have carefully read the ICLR Code of Ethics. We are sure that all studies and procedures described in the paper follow the ethical rules in the academia. Our research does not involve human subjects, and also does not contain any potentially harmful insights, methodologies and applications, Furthermore, we do not expect any direct discrimination/bias/fairness concerns, or privacy/security issues relating to this paper.

**Reproducibility Statement**

All key technical details and quantitative results are provided in the main paper and Appendix, enabling researchers and practitioners in this field to reproduce our work. Additionally, all datasets used in this research are publicly available to the community. We will also release the source code of our method upon acceptance of the paper.

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

## A  APPENDIX

### A.1  FURTHER IMPLEMENTATION DETAILS

We complement the implementation details shared in Subsection 4.1: our solution is based on the official implementations of 3DGS Kerbl et al. (2023) and LangSplat Qin et al. (2024) made publicly available by their respective authors (*c.f.* links provided in their papers). Specifically, we edit the LangSplat's 3DGS model to add our novel smoothed indicator, as well as the CUDA rasterization function for joint rendering of color and semantic modalities w.r.t. their respective opacity/indicator control. As explained in the main paper, we also adapt the optimization script for joint learning over both modalities and for integrating the proposed view-blending augmentation, applied to pairs of batched input samples at every iteration. The learning rates applied to the different 3DGS attributes are provided in Tab. 7.

Table 7: Learning rates applied to the parameters of the Gaussians.

| Components | Learning Rates |
|---|---|
| position | $1.6 \times 10^{-4}$ |
| scale | $5.0 \times 10^{-3}$ |
| rotation | $1.0 \times 10^{-3}$ |
| color features | $5.0 \times 10^{-3}$ |
| color opacity | $5.0 \times 10^{-2}$ |
| semantic features | $5.0 \times 10^{-3}$ |
| semantic indicator | $5.0 \times 10^{-2}$ |

For a larger version of Sub-figs. 2.C-D representing our fusion module and augmentation module respectively, we refer the readers to Fig. 9.

In the ablation on cross-modal rasterizer (Sec. 4.3), the cross-attention fusion tested in Tab. 4.A is performed by using $f^i$ as query and $c^i$ as both key and value, whereas the adopted self-attention performs cross-modality fusion by using $c^i \oplus f^i$ as query, key, and value.

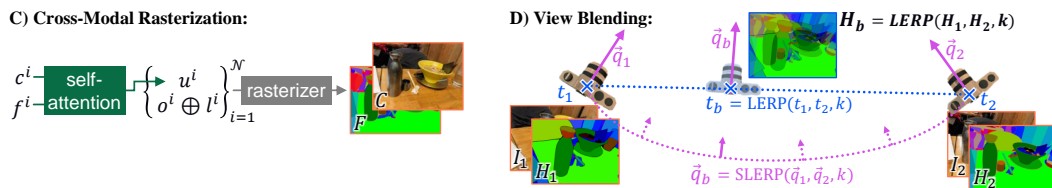

Figure 9: Larger version of Sub-figs. 2.C-D describing our fusion and augmentation modules.

Table 8: mIoU results on the Mip-NeRF 360 dataset.

| Method | Venue | garden | bonsai | kitchen | room | avg. |
|---|---|---|---|---|---|---|
| Feature-3DGS | CVPR'24 | 45.1 | 46.2 | 46.8 | 17.5 | 38.9 |
| LEGaussians | CVPR'24 | 48.5 | 50.4 | 49.3 | 55.7 | 51.0 |
| LangSplat | CVPR'24 | 50.1 | 59.1 | 50.0 | 62.6 | 55.5 |
| GS-Grouping | ECCV'24 | 42.0 | 43.1 | 42.2 | 49.1 | 44.1 |
| GOI | ACMMM'24 | 85.0 | 91.5 | 84.3 | 85.0 | 86.5 |
| **Ours** | | **89.2** | **93.7** | **87.6** | **88.8** | **89.8** |

## A.2 EVALUATION ON ADDITIONAL MIP-NERF 360 DATASET

### A.2.1 QUANTITATIVE RESULTS.

Building on the work of GOI (Qu et al., 2024), we also utilize the Mip-NeRF 360 dataset (Barron et al., 2022) for our evaluation. In this context, we employ APE (Shen et al., 2024) as the frozen backbone to extract ground truth 2D features. The results, presented in Tab. 8, clearly demonstrate the effectiveness of our proposed method, highlighting its ability to improve overall performance on this dataset.

### A.2.2 QUALITATIVE RESULTS.

In Fig. 10 and Fig. 11, we present qualitative comparisons on the Mip-NeRF 360 dataset (Barron et al., 2022). The results demonstrate that our proposed method significantly outperforms the competing approaches.

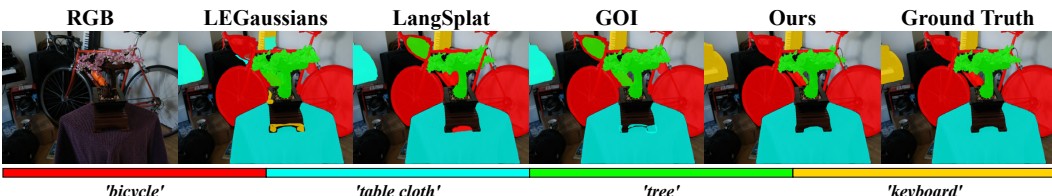

Figure 10: Qualitative semantic segmentation comparisons on the `bonsai` scene of Mip-NeRF 360.

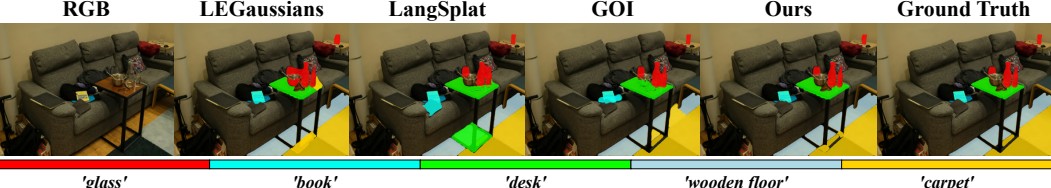

Figure 11: Qualitative semantic segmentation comparisons on the `room` scene of Mip-NeRF 360.

## A.3 ADDITIONAL ABLATION STUDIES

### A.3.1 HIGH-LEVEL ABLATION OF PROPOSED CONTRIBUTIONS.

We first provide some high-level ablation studies Tab. 9 and Tab. 10, highlighting the impact of each of our key contributions, *i.e.*, our proposed modality fusion (" modal. fus."), semantic indicator ("

Table 9: Ablation results of three key contributions on the LERF dataset, in terms of mIoU.

| modal. fus. | sem. indic. | view blend. | ramen | teatime | figurines | kitchen | avg. |
|:-:|:-:|:-:|:-:|:-:|:-:|:-:|:-:|
| × | × | × | 53.3 | 66.9 | 46.3 | 45.3 | 53.0 |
| × | × | ✓ | 54.7 | 68.5 | 47.7 | 45.8 | 54.2 |
| × | ✓ | × | 55.4 | 67.2 | 47.8 | 48.9 | 54.8 |
| ✓ | × | × | 54.8 | 67.5 | 48.2 | 48.6 | 54.8 |
| × | ✓ | ✓ | 57.8 | 69.6 | 54.2 | 51.5 | 58.3 |
| ✓ | × | ✓ | 55.9 | 71.0 | 54.2 | 51.3 | 58.1 |
| ✓ | ✓ | × | 55.4 | 68.2 | 48.8 | 50.9 | 55.8 |
| ✓ | ✓ | ✓ | **61.4** | **73.5** | **58.1** | **54.8** | **62.0** |

Table 10: Ablation results of three key contributions on the 3D-OVS dataset, in terms of mIoU.

| modal. fus. | sem. indic. | view blend. | bed | bench | room | sofa | lawn | avg. |
|:-:|:-:|:-:|:-:|:-:|:-:|:-:|:-:|:-:|
| × | × | × | 92.7 | 94.3 | 94.1 | 90.5 | 96.0 | 93.5 |
| × | × | ✓ | 93.0 | 94.6 | 94.9 | 91.6 | 96.1 | 94.0 |
| × | ✓ | × | 93.6 | 95.1 | 95.0 | 92.2 | 96.0 | 94.4 |
| ✓ | × | × | 93.9 | 95.2 | 95.3 | 91.8 | 96.2 | 94.5 |
| × | ✓ | ✓ | 94.5 | 95.6 | 95.3 | 93.0 | 96.5 | 95.0 |
| ✓ | × | ✓ | 94.7 | 95.2 | 95.0 | 92.7 | 96.4 | 94.8 |
| ✓ | ✓ | × | 92.9 | 94.6 | 95.2 | 91.5 | 96.1 | 94.1 |
| ✓ | ✓ | ✓ | **96.8** | **97.3** | **97.7** | **95.5** | **97.9** | **97.1** |

Table 11: Ablation results of modality fusion on the 3D-OVS dataset.

| Method | bed | bench | room | sofa | lawn | avg. |
|:-:|:-:|:-:|:-:|:-:|:-:|:-:|
| No Fusion | 94.5 | 95.6 | 95.3 | 93.0 | 96.5 | 95.0 |
| Single-layer MLP Fusion | 93.1 | 94.7 | 94.4 | 91.7 | 96.0 | 94.1 |
| Cross-attention Modality Fusion | 94.7 | 95.9 | 95.6 | 93.2 | 96.6 | 95.2 |
| Self-attention Modality Fusion (Ours) | **96.8** | **97.3** | **97.7** | **95.5** | **97.9** | **97.1** |

Table 12: Ablation results of smoothed semantic indicator on the 3D-OVS dataset.

| Method | bed | bench | room | sofa | lawn | avg. |
|:-:|:-:|:-:|:-:|:-:|:-:|:-:|
| Fixed to 0.5 | 93.4 | 94.9 | 95.1 | 92.0 | 95.4 | 94.2 |
| Fixed to 1.0 | 91.5 | 92.8 | 90.9 | 87.6 | 92.8 | 91.1 |
| Using Color Opacity | 94.7 | 95.2 | 95.0 | 92.7 | 96.4 | 94.8 |
| Smoothed Semantic Indicator (Ours) | **96.8** | **97.3** | **97.7** | **95.5** | **97.9** | **97.1** |

sem. indic."), and camera-view blending ("view blend.") . The results demonstrate that all three components are crucial in improving the overall performance.

### A.3.2 ABLATION OF MODALITY FUSION

Table 11 presents an ablation study of modality fusion using the OVS-3D dataset. The results indicate that self-attention modality fusion is the most effective method.

### A.3.3 ABLATION OF SEMANTIC INDICATOR

Tab. 12 compares our proposed Smoothed Semantic Indicator against three alternatives: Fixed Value at 0.5, Fixed Value at 1, and Using Color Opacity on the 3D-OVS dataset. The results demonstrate that fixed values greatly under-perform compared to shared opacity derived from color rendering. Moreover, the learnable Smoothed Semantic Indicator surpasses the use of color opacity significantly. Thus, utilizing a learnable semantic indicator for semantic rasterization proves beneficial for the open-vocabulary segmentation task.

Table 13: Comparison between data-augmentation strategies, over downstream open-vocabulary semantic segmentation (mIoU) on LERF scenes. Note that the original training sets contain between 124 and 297 images per scene. For the offline augmentation using 3DGS-rendered images and off-the-shelf open-vocabulary models (SAM/CLIP), 120 novel views are added, with their viewpoints sampled via interpolation.

| Method | ramen | teatime | figurines | kitchen | avg. |
|---|---|---|---|---|---|
| w/o data augmentation | 53.3 | 66.9 | 46.3 | 45.3 | 53.0 |
| w/ data from 3DGS & SAM/CLIP | 55.2 | 68.1 | 49.5 | 48.2 | 55.3 |
| w/ view blending | **61.4** | **73.5** | **58.1** | **54.8** | **62.0** |

Table 14: Ablation results of camera view blending on the 3D-OVS dataset.

| Rotation | Translation | SSIM | bed | bench | room | sofa | lawn | avg. |
|---|---|---|---|---|---|---|---|---|
| ✗ | ✗ | ✗ | 92.9 | 94.6 | 95.2 | 91.5 | 96.1 | 94.1 |
| ✗ | ✓ | ✗ | 93.5 | 94.9 | 95.5 | 92.2 | 96.2 | 94.5 |
| ✗ | ✓ | ✓ | 94.8 | 95.2 | 96.3 | 93.5 | 96.4 | 95.2 |
| ✓ | ✗ | ✗ | 94.0 | 94.8 | 95.7 | 92.3 | 96.3 | 94.6 |
| ✓ | ✗ | ✓ | 95.1 | 95.4 | 96.2 | 93.0 | 96.2 | 95.2 |
| ✓ | ✓ | ✗ | 96.2 | 96.8 | 96.7 | 94.6 | 96.4 | 96.1 |
| ✓ | ✓ | ✓ | **96.8** | **97.3** | **97.7** | **95.5** | **97.9** | **97.1** |

Table 15: Ablation results of interpolation ratio sampling on the 3D-OVS dataset.

| Method | bed | bench | room | sofa | lawn | avg. |
|---|---|---|---|---|---|---|
| Fixed to 0.5 | 95.8 | 95.9 | 95.6 | 93.7 | 96.5 | 95.5 |
| Fixed to 0.75 | 94.1 | 95.2 | 95.0 | 92.5 | 95.4 | 94.4 |
| Fixed to 1.0 | 93.4 | 94.0 | 94.4 | 90.8 | 95.2 | 93.6 |
| Even Distribution | 94.6 | 95.1 | 94.8 | 93.2 | 96.1 | 94.8 |
| Gaussian Distribution | 94.0 | 94.5 | 94.2 | 92.9 | 96.0 | 94.3 |
| Beta Distribution (Ours) | **96.8** | **97.3** | **97.7** | **95.5** | **97.9** | **97.1** |

Table 16: Ablation results of the parameters of Beta distribution on LERF data, in terms of mIoU.

| Method | ramen | teatime | figurines | kitchen | avg. |
|---|---|---|---|---|---|
| Beta(0.1, 0.1) | 59.7 | 72.2 | 57.3 | 54.4 | 60.9 |
| Beta(0.2, 0.2) | **61.4** | **73.5** | **58.1** | **54.8** | **62.0** |
| Beta(0.3, 0.3) | 60.7 | 72.8 | 57.1 | 54.2 | 61.2 |
| Beta(0.4, 0.4) | 59.6 | 72.0 | 56.8 | 54.1 | 60.6 |

### A.3.4 ABLATION OF CAMERA VIEW BLENDING

We first provide further justifications for our view-blending augmentation technique. Instead of our mix-up strategy, one could apply the selected off-the-shelf language-feature extractor (*e.g.*, CLIP) to generate new pseudo-ground-truth data. This could be done either:

- *online*, *i.e.*, using the model's predicted color image as input to the language-feature extractor in order to generate the target language map to evaluate the one rendered by the model.
- *offline*, *i.e.*, using a pre-trained color-only 3DGS model of the scene to generate novel views, pass these views to the feature extractor, and use all the resulting data to train the language-embedded representation.

However, the computational cost of running the selected language-feature extractor (CLIP) is too heavy for online usage, and the offline strategy also comes with several downsides.

First, one of our contributions is the joint CUDA-based rasterization function, which enables the fast rendering of both modalities (color and language). While most prior works Qin et al. (2024) train their model in two phases (a first phase to train a color-only 3DGS model, then a second phase to extend it with language information), we leverage our custom rasterizer to efficiently optimize our vision-language model in a single training phase (see Table 8 highlighting our training and rendering

Table 17: Ablation results on different levels of disentanglement between the per-modality Gaussian parameters, evaluated on the downstream open-vocabulary semantic-segmentation task on LERF.

| | Parameters shared across modalities | | | | Segmentation accuracy (mIoU) | | | | |
| | position | covariance | opacity/indic. | features | ramen | teatime | figurines | kitchen | avg. |
|---|---|---|---|---|---|---|---|---|---|
| | ✓ | ✓ | ✓ | × | 55.9 | 71.0 | 54.2 | 51.3 | 58.1 |
| (ours) | ✓ | ✓ | × | × | **61.4** | **73.5** | **58.1** | **54.8** | **62.0** |
| | ✓ | × | × | × | 47.8 | 65.6 | 42.3 | 40.9 | 49.2 |
| | × | × | × | × | 41.4 | 60.7 | 39.2 | 38.8 | 45.0 |

efficiency). The downside of this faster training is that, unlike prior solutions, we do not have access to a pre-trained 3DGS model of the target scene. Adding a pre-training phase for the color-only model would be a significant computational overhead. Second, the aforementioned strategy (*i.e.*, rendering additional views using a pre-trained 3DGS) only adds a fixed/limited number of images to the training set (*c.f.* *offline* rendering before the training starts), whereas our augmentation scheme provides new randomized language maps at every training iteration (*c.f.* *online* augmentation).

We quantitatively compared the suggested offline augmentation strategy to ours in a new experiment shared in Tab. 13, which confirms that our augmentation technique does indeed result in a more robust representation w.r.t. downstream semantic tasks. Note that in future work, it would be interesting to investigate how both strategies could be optimally interleaved (*e.g.*, by pretraining our model with mix-up augmentation; then, after some iterations, augmenting the training data with 3DGS/CLIP-rendered pairs).

Additionally, we present an ablation study of various camera view blending strategies in Tab. 14 using the 3D-OVS dataset, which includes three variants: Rotation, Translation, and SSIM. Our proposed method is represented in the last row of the table. The results indicate that both Slerp-based rotation blending and Lerp-based translation blending enhance performance. Furthermore, the regularization provided by SSIM is crucial for managing the extent of blending.

In Tab. 15, we further present an ablation study on the interpolation ratio used within the camera view blending scheme, performed on 3D-OVS data. Alongside fixed values of $0.5$, $0.75$, and $1.0$, we also include experiments utilizing Even Distribution ($U(0, 1)$), Gaussian Distribution ($N(0, 1)$), and our Beta Distribution ($Beta(0.2, 0.2)$). The results indicate that a balanced ratio is essential for achieving better representations, and that minor disturbances can also yield beneficial effects.

We also change the parameter of the Beta distribution in the blending loss $\mathcal{L}_b$ on the LERF dataset, and the results are exhibited in Tab. 16, showing our selection is reasonable. It is important to keep a relatively balanced ratio while introducing some disturbances.

### A.3.5 Ablation on Disentanglement Level between the Two Modalities

The main reason why prior 3DGS-based or NeRF-based language-embedded scene representations subordinate the semantic modality to the visual one is that the 2D semantic maps are too sparse to regress the underlying 3D scene geometry from them. The visual/shading information provided by the corresponding color images is required for the 3D representation to properly converge. Moreover, for some downstream tasks such as 3D editing, it is necessary to maintain the same underlying geometry across modalities. For these reasons, we decided to keep all spatial properties of the 3D Gaussians (*i.e.*, mean position and covariance) common to the two modalities, and to disentangle only the modality-specific parameters (color features vs. language features, opacity vs. semantic indicator).

We substantiate the above with an additional experiment, shared in Tab. 17. We evaluate multiple variations of our solution over the open-vocabulary semantic segmentation task, with different levels of disentanglement between the represented modalities: (1) a model similar to prior work, with only the feature vectors specific to each modality; (2) our proposed solution, introducing per-modality opacity/indicator; (3) a model further introducing per-modality covariance matrices; (4) a model with completely separate Gaussians for each modality (similar to training a semantic-only 3DGS model for the segmentation task). As expected, the semantic-only model (4) fails to effectively learn the underlying 3D information and converges poorly, impacting the downstream results on unseen views. Our solution performs the best out of the four.

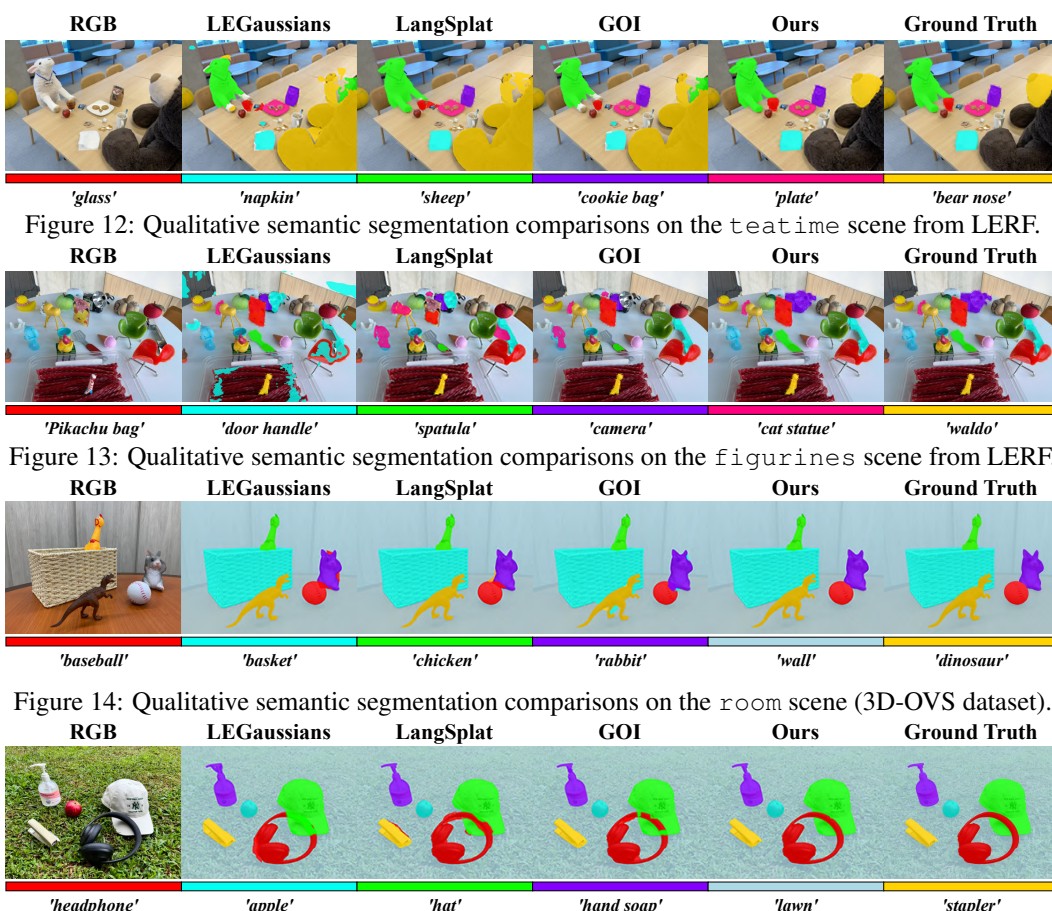

Figure 12: Qualitative semantic segmentation comparisons on the `teatime` scene from LERF.

Figure 13: Qualitative semantic segmentation comparisons on the `figurines` scene from LERF.

Figure 14: Qualitative semantic segmentation comparisons on the `room` scene (3D-OVS dataset).

Figure 15: Qualitative semantic segmentation comparisons on the `lawn` scene (3D-OVS dataset).

## A.4 ADDITIONAL QUALITATIVE RESULTS

### A.4.1 RESULTS ON LERF DATA

Fig. 12 and Fig. 13 present additional semantic segmentation results on the LERF dataset, while Fig. 16 displays further localization results for the same dataset. Our proposed method clearly outperforms other approaches, particularly in reflective and translucent areas, where it more closely aligns with the ground truth.

### A.4.2 RESULTS ON 3D-OVS DATA

We present additional segmentation comparisons on the 3D-OVS dataset in Fig. 14 and Fig. 15. Our proposed method demonstrates superior performance compared to other approaches, showing greater alignment with the ground truth.

### A.4.3 IMPACT OF SEMANTIC-INDICATOR CONTRIBUTION ON RELEVANCY MAPS

In Fig. 17, we share the relevancy maps (represented as heatmaps) corresponding to various open-vocabulary prompts, comparing the results for color/semantic 3DGS models incorporating our novel smoothed semantic (last row) versus baseline models re-using the color opacity during rasterization of semantic data (mid row). The results clearly demonstrate the effectiveness of the semantic indicator in uncovering semantic information, particularly in translucent and reflective objects. This enhanced capability allows for a more nuanced understanding of the scene, highlighting how the proposed method improves the semantic representations.

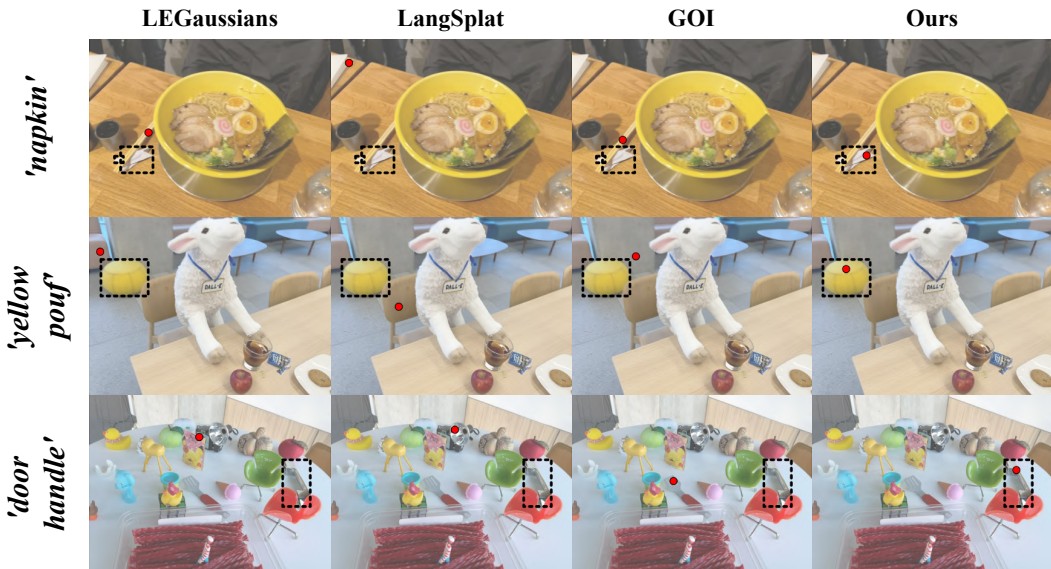

Figure 16: Additional qualitative examples of object localization in the LERF dataset.

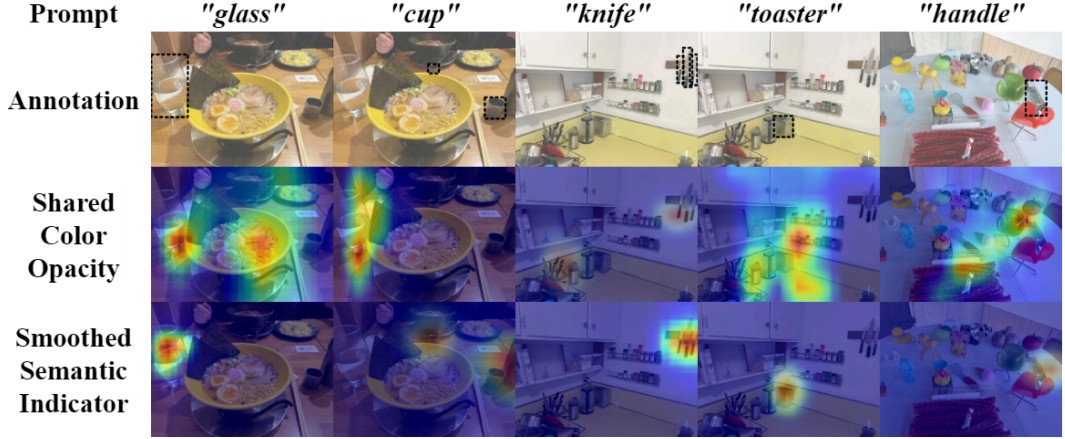

Figure 17: Qualitative impact of the proposed additional semantic indicator on 2D relevancy maps w.r.t. various open-vocabulary prompts.

### A.4.4  IMPACT OF CAMERA-VIEW BLENDING ON COLOR/LANGUAGE DISENTANGLEMENT

In Figs. 18 and 19, we demonstrate how camera-view blending helps alleviate over-fitting. Fig. 18 shows examples of multiple objects sharing the same colors. For example, in the first scene, the scissors and spatula share the same color, leading part of the scissors to be misclassified as spatula when camera-view blending is not applied. Similarly, in the second scene, the pot and table share the same wooden texture, causing part of the pot to be misidentified as table by the model without camera-view blending.

Fig. 19 shows examples of objects composed of multiple colors/textures. In the first example, the camera and the spatula are composed of parts with different colors, and some of these parts end up misclassified by the model without camera-view blending. The same can be observed in the second example: the bike is composed of parts with varied appearances, and the model optimized without our augmentation misses many of them; whereas the model optimized with correctly identifies the whole object. The proposed technique enhances semantic consistency across diverse views through regularization.

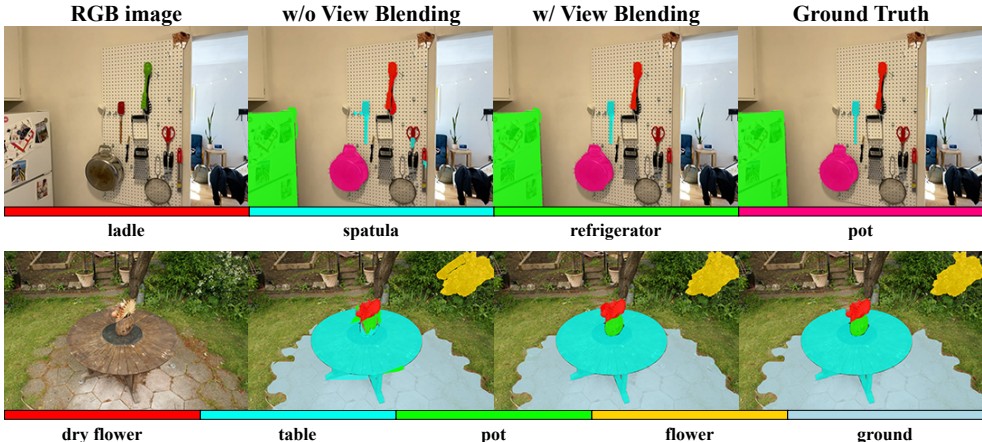

Figure 18: Examples of multiple objects sharing similar colors/textures (*e.g.*, scissors and spatula in example #1, pot and table in example #2).

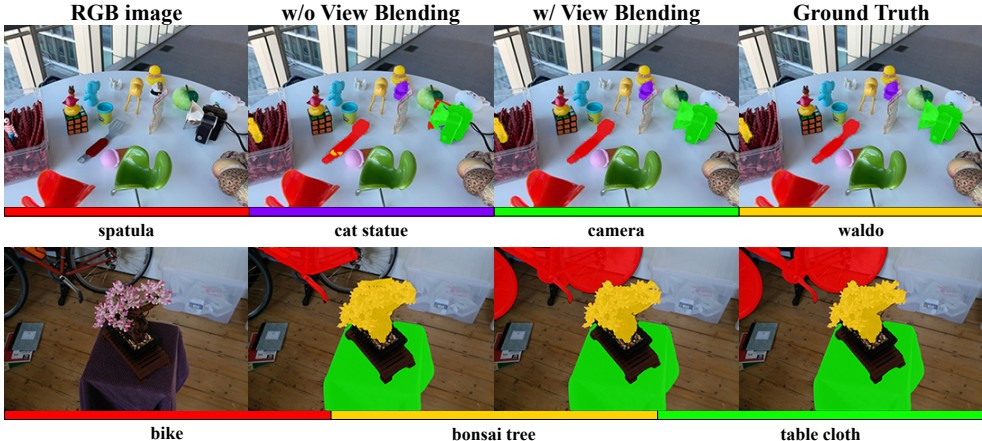

Figure 19: Examples of objects composed of multiple colors/textures (*e.g.*, multi-part camera and spatula in example #1, bike in example #2).

## A.5 ADDITIONAL QUANTITATIVE RESULTS

### A.5.1 EXTENDED RESULTS ON 3D-OVS SCENES

In the main manuscript, we considered the five main scenes of the 3D-OVS dataset, following the protocol adopted in LangSplat Qin et al. (2024). To match other experimental protocols, such as FMGS' one which include the additional `table` scene, we extend our evaluation to additional 3D-OVS scenes, for a total of seven, *i.e.*, adding results for `table` and `snacks` scenes. The results shared in Tab. 18 confirm the effectiveness of our solution in terms of open-vocabulary semantic segmentation.

Table 18: Evaluation on 3D-OVS dataset extended to additional scenes.

| Method | Venue | bed | bench | room | sofa | lawn | table | snacks | avg. |
|---|---|---|---|---|---|---|---|---|---|
| Feature-3DGS | CVPR'24 | 83.5 | 90.7 | 84.7 | 86.9 | 93.4 | 92.4 | 88.6 | 88.6 |
| LEGaussians | CVPR'24 | 84.9 | 91.1 | 86.0 | 87.8 | 92.5 | 93.6 | 87.5 | 89.1 |
| LangSplat | CVPR'24 | 92.5 | 94.2 | 94.1 | 90.0 | 96.1 | 95.9 | 92.3 | 93.6 |
| GS-Grouping | ECCV'24 | 83.0 | 91.5 | 85.9 | 87.3 | 90.6 | 94.2 | 88.7 | 88.8 |
| GOI | ACMMM'24 | 89.4 | 92.8 | 91.3 | 85.6 | 94.1 | 95.6 | 89.3 | 91.1 |
| FMGS | IJCV'24 | 80.6 | 84.5 | 87.9 | 90.8 | 92.6 | 97.2 | 83.0 | 88.2 |
| Ours | | **96.8** | **97.3** | **97.7** | **95.5** | **97.9** | **98.5** | **94.7** | **97.0** |

A.5.2    EXTENDED EFFICIENCY ANALYSIS AND IMAGE QUALITY EVALUATION

We complement the Tab. 6 from the main paper, adding results on other LERF scenes. As shown in Tabs. 19-22, our proposed method consistently outperforms others in terms of the training time, FPS, number of Gaussians, and storage size.

Furthermore, we extend these experiments to measure the quality of the color images synthesized by the trained Gaussian splatting models. The goal is to verify that, while this work focuses on semantic accuracy, our solution does not sacrifice visual precision too much. PSNR and SSIM results in Tab. 23 actually show that our model remains competitive on the visual modality, showing the effectiveness of our modality balance. Moreover, comparing to the results from color-only 3DGS (same as LangSplat as this method fixes all 3DGS parameters after its pre-training), we observe that semantic learning does not have an obvious impact on color-modality metrics (*i.e.*, image quality).

Table 19: Efficiency and image-quality analysis on LERF's `ramen`.

| Method | Training Time ↓ | FPS ↑ | Gaussians # ↓ | Storage Size ↓ | PSNR ↑ |
|---|---|---|---|---|---|
| LangSplat | 96min | 40 | 86k | 180MB | 28.5 |
| GS-Grouping | 130min | 76 | 107k | 230MB | **29.2** |
| GOI | 73min | 42 | 92k | 198MB | 29.0 |
| Ours | **65min** | **79** | **80k** | **175MB** | **29.2** |

Table 20: Efficiency and image-quality analysis on LERF's `teatime`.

| Method | Training Time ↓ | FPS ↑ | Gaussians # ↓ | Storage Size ↓ | PSNR ↑ |
|---|---|---|---|---|---|
| LangSplat | 102min | 42 | 152k | 433MB | **32.6** |
| GS-Grouping | 135min | 75 | 187k | 506MB | 32.1 |
| GOI | 79min | 44 | 170k | 467MB | 32.3 |
| Ours | **67min** | **78** | **144k** | **425MB** | 32.5 |

Table 21: Efficiency and image-quality analysis on LERF's `figurines`.

| Method | Training Time ↓ | FPS ↑ | Gaussians # ↓ | Storage Size ↓ | PSNR ↑ |
|---|---|---|---|---|---|
| LangSplat | 121min | 38 | 94k | 190MB | **27.9** |
| GS-Grouping | 130min | 65 | 116k | 245MB | 26.5 |
| GOI | 104min | 39 | 92k | 222MB | 27.7 |
| Ours | **89min** | **67** | **86k** | **187MB** | **27.9** |

A.5.3    PER-CATEGORY EVALUATION OF SEMANTIC INDICATOR CONTRIBUTION

In the following, we expand on our justification for introducing the semantic indicator $l$ provided in Sec. 3.2, especially Fig. 3. There, we highlight how re-using the opacity $o$ (optimized w.r.t. the color modality) to rasterize language maps—*i.e.*, entangling visual properties and 2D language projections—can harm the semantic accuracy for objects with complex visual properties, such as translucent or highly-reflective objects. Here, we provide further empirical evidence for disentangling the rasterization properties of the two modalities, especially for translucent/reflective object categories. In Tab. 24 through Tab. 28, we present the mIoU results for each ground-truth category, highlighting objects considered translucent or reflective (*e.g.*, glass, metal bottle, *etc.*). We observe how the introduction of the semantic indicator specially benefits these categories.

In other words: as shown in Fig. 3, translucent and reflective objects are mostly represented by 3D Gaussians with low opacity values (in order to model their complex light transport). Using these low opacity values to rasterize the language features cause unwanted artifacts. This is alleviated by our semantic indicator. Our model correctly learns higher indicator values for Gaussians representing the above-mentioned object categories (*e.g.*, see large $l - o$ results for the translucent glass or reflective bottle/cup in Fig. 3). The improvement on downstream semantic tasks, *e.g.*, open-vocabulary semantic segmentation, is thus especially large for these categories. *E.g.*, the segmentation of `glass`

Table 22: Efficiency and image-quality analysis on LERF's `kitchen`.

| Method | Training Time ↓ | FPS ↑ | Gaussians # ↓ | Storage Size ↓ | PSNR ↑ |
|---|---|---|---|---|---|
| LangSplat | 105min | 43 | 142k | 406MB | **32.4** |
| GS-Grouping | 141min | 77 | 177k | 455MB | 32.3 |
| GOI | 86min | 45 | 162k | 429MB | 31.9 |
| Ours | **80min** | **78** | **135k** | **398MB** | 32.3 |

Table 23: Evaluation of color-image rendering quality on the LERF dataset.

| Method | ramen | | teatime | | figurines | | kitchen | | avg. | |
|---|---|---|---|---|---|---|---|---|---|---|
| | PSNR | SSIM | PSNR | SSIM | PSNR | SSIM | PSNR | SSIM | PSNR | SSIM |
| LangSplat (3DGS) | 28.5 | **0.87** | **32.6** | **0.83** | **27.9** | 0.76 | **32.4** | 0.85 | 30.4 | **0.83** |
| GS-Grouping | **29.2** | 0.85 | 32.1 | 0.82 | 26.5 | 0.74 | 32.3 | 0.84 | 30.0 | 0.81 |
| GOI | 29.0 | 0.83 | 32.3 | 0.83 | 27.7 | **0.77** | 31.9 | 0.82 | 30.2 | 0.81 |
| Ours | **29.2** | 0.86 | 32.5 | **0.83** | **27.9** | 0.76 | 32.3 | **0.86** | **30.5** | **0.83** |

is improved by 14.5% and `sake_cup` by 14.3%, whereas the segmentation of non-translucent/non-reflective is improved by 3.6% on average, *c.f.* Tab. 24.

Table 24: Per-category mIoU results for the open-vocabulary semantic segmentation task on LERF's `ramen` scene.

| Semantic Indicator? | *non-translucent and non-reflective object categories* | | | | | | | | *object categories with translucent/reflective parts* | | | | |
| --- | --- | --- | --- | --- | --- | --- | --- | --- | --- | --- | --- | --- | --- |
| | chopsticks | egg | nori | napkin | noodles | kamaboko | onion.segments | corn | bowl | sake.cup | plate | glass | spoon |
| × | 90.0 | 96.4 | 78.2 | 45.9 | 51.4 | 37.7 | 24.7 | 10.4 | 40.3 | 36.7 | 53.6 | 64.1 | 30.3 |
| ✓ | 90.4 | 96.5 | 80.4 | 51.2 | 56.6 | 44.5 | 29.8 | 13.9 | 53.7 | 51.0 | 62.1 | 78.6 | 45.4 |
| improvement | +0.4 | +0.1 | +2.2 | +5.3 | +5.2 | +6.8 | +5.1 | +3.5 | +13.4 | +14.3 | +8.5 | +14.5 | +15.1 |

Table 25: Per-category mIoU results for the open-vocabulary semantic segmentation task on LERF's `teatime` scene.

| Semantic Indicator? | *non-translucent and non-reflective object categories* | | | | | | | | | *translucent/reflective objects* | | | | |
| --- | --- | --- | --- | --- | --- | --- | --- | --- | --- | --- | --- | --- | --- | --- |
| | bear | cookie.bag | sheep | apple | napkin | bear.nose | cookies | coffee | hooves | brand | pouf | coffee.mug | plate | glass |
| × | 62.7 | 71.5 | 85.8 | 77.1 | 64.3 | 24.9 | 76.0 | 46.9 | 14.6 | 52.7 | 53.2 | 79.1 | 88.6 | 64.7 |
| ✓ | 66.9 | 72.2 | 85.9 | 77.6 | 68.1 | 40.3 | 76.5 | 51.2 | 19.8 | 58.1 | 55.6 | 88.6 | 95.4 | 77.6 |
| improvement | +4.2 | +0.7 | +0.1 | +0.5 | +3.8 | +15.4 | +0.5 | +4.3 | +5.2 | +5.4 | +2.4 | +9.5 | +6.8 | +12.9 |

Table 26: Per-category mIoU results for the open-vocabulary semantic segmentation task on LERF's `kitchen` scene.

| Semantic Indicator? | *non-translucent and non-reflective object categories* | | | | | | | | *object categories with translucent/reflective parts* | | | |
| --- | --- | --- | --- | --- | --- | --- | --- | --- | --- | --- | --- | --- |
| | desk | ottolenghi | refrigerator | ketchup | cabinet | plate | knife | toaster | stainless.pots | vessel | ladle | pot |
| × | 39.3 | 60.7 | 50.7 | 20.5 | 29.6 | 69.4 | 46.0 | 24.8 | 42.3 | 28.7 | 17.0 | 68.8 |
| ✓ | 40.0 | 62.1 | 55.3 | 23.2 | 33.7 | 78.8 | 52.5 | 45.6 | 51.5 | 44.2 | 32.4 | 76.5 |
| improvement | +0.7 | +1.4 | +4.6 | +2.7 | +4.1 | +9.4 | +6.5 | +20.8 | +9.2 | +15.5 | +15.4 | +7.7 |

Table 27: Per-category mIoU results for the open-vocabulary semantic segmentation task on LERF's `figurines` scene (part 1/2).

| Semantic Indicator? | `elephant` | `waldo` | `hat_duck` | `buoy_duck` | *non-translucent and non-reflective object categories* | | | | | | | | |
| --- | --- | --- | --- | --- | --- | --- | --- | --- | --- | --- | --- | --- | --- |
| | | | | | `ice_cream` | `green_apple` | `pikachu` | `red_apple` | `jake` | `hat` | `magic_cube` | `pumpkin` | |
| × | 90.0 | 78.3 | 56.8 | 54.4 | 42.2 | 84.2 | 37.6 | 64.5 | 42.8 | 21.2 | 74.8 | 73.3 | ⋯ |
| ✓ | 90.0 | 80.5 | 59.2 | 58.7 | 45.6 | 84.4 | 44.0 | 67.8 | 43.3 | 24.7 | 76.2 | 75.1 | |
| improvement | +0.0 | +2.2 | +2.4 | +4.3 | +3.4 | +0.2 | +6.4 | +3.3 | +0.5 | +3.5 | +1.4 | +1.8 | |

Table 28: Per-category mIoU results for the open-vocabulary semantic segmentation task on LERF's `figurines` scene (part 2/2).

| Semantic Indicator? | `camera` | `tesla_handle` | *object categories with translucent/reflective parts* | | | | |
| --- | --- | --- | --- | --- | --- | --- | --- |
| | | | `porcelain_hand` | `red_chair` | `spatula` | `cat_statue` | `green_chair` |
| × | 49.6 | 28.2 | 41.9 | 48.6 | 45.3 | 37.5 | 64.8 |
| ✓ | 61.3 | 43.7 | 50.4 | 65.4 | 60.6 | 51.0 | 76.6 |
| improvement | +11.7 | +15.5 | +8.5 | +16.8 | +15.3 | +13.5 | +11.8 |

