# OpenReview forum: "3D Vision-Language Gaussian Splatting"
_ICLR.cc/2025/Conference — ICLR 2025 Poster_

### Official Review · Reviewer_56RB · 2024-10-19

**Soundness:** 3
**Presentation:** 3
**Contribution:** 3
**Rating:** 6
**Confidence:** 3

**Summary:**

This paper proposes a novel approach to open-vocabulary 3DGS. It proposes to use another learnable opacity parameter for the blending of semantics, and train a cross-modal fusion module. The authors also propose a semantic-aware camera view blending method, in which they randomly interpolate between training views and supervise the semantic rendering based on the weighted sum of the semantics. The experiments show that the proposed method significantly improves the open-vocabulary semantic segmentation results.

**Strengths:**

1. The proposed method is novel and promising. Using another learnable parameter as opacity for semantic blending is an overlooked method that benefits open-vocabulary 3DGS.
2. The results are promising.

**Weaknesses:**

Please see the questions below.

**Questions:**

1. Is the fusion module per-scene trained? Would it be more effective to train it across different scenes to learn the prior?
2. According to Table 6 and Table 7, does the view interpolation method alone improve the performance by 6.2 mAP? It's a little counter-intuitive that simply weighted-summing the two views to supervise the interpolated views can lead to such improvements, since the weighted-sum semantics (as shown in Figure 2) do not consider the relative camera poses between the two views. What if you wrap the two views to the interpolated view using the known camera poses and learned depths?
3. The paper discussed using separate opacity for semantics, then what would the results be if learning separate covariance matrix in each gaussian for color and semantics? Besides, if we simply train two separate sets of Gaussians for color and semantic renderings respectively, would it improve the semantic segmentation results?

---

> ### Author Response · Authors · 2024-11-21
>
> ## Q1. Per-scene training of fusion module
>
> The reviewer is correct that the fusion module is trained for each scene. We followed the protocol from prior works (LangSplat, GOI) in optimizing our framework separately for each scene. We agree with the reviewer that the generalization of language-embedded Gaussian representations, or at least some of their components (e.g., fusion module), is an interesting open challenge; but we believe that it falls beyond the scope of our current work.
>
> E.g., an obstacle towards generalizing our fusion module across scenes is its dependency on the feature-compression model (since the fusion module receives compressed language features). This compression model is itself specific to each new scene (in LangSplat and our work, we use an auto-encoder trained on the subset of scene-specific language embeddings; but other works also use scene-specific compression techniques, such as codebooks in GOI). Generalizing the compression model would come with a significant computational cost for the whole framework: current compression techniques are extremely effective _because_ they are scene-specific. E.g., our auto-encoder can reduce CLIP vectors from 768 dimensions to only 3 simply because the number of unique language concepts relevant to a single scene is relatively small.
>
> A less effective compression would mean a higher dimensionality for the Gaussians (c.f. larger 3D language vector), impacting everything from inference time to model optimization. Generalizing components of language-embedded Gaussian-based frameworks is, therefore, a challenging task, deserving of its own research.
>
>
> ----
> ## Q2. View interpolation
>
> To be candid, we were also pleasantly surprised by the effectiveness of the proposed view-blending augmentation, confirming our intuition w.r.t. the impact of cross-view regularization over the sparse language modality. For an additional discussion on augmentation strategies, we invite the reviewer to check our response **W2** to Reviewer `Awye`.
>
> As mentioned there, we could use the color results of the 3DGS model to guide the augmentation process, either in an online or offline manner. Similarly, we could also, indeed, extend the 3DGS model to render depth images, to help reproject/wrap language maps to new viewpoints. However, such a depth-based strategy comes with similar downsides as the one proposed by Reviewer `Awye`:
> - As an _online_ augmentation:
>     - Depth-based image wrapping is a computationally-heavy operation compared to 2D interpolation, so its use as online augmentation may be unrealistic for many setups.
>     - Adding the online rendering of depth maps by the Gaussian model would further slow down the training.
> - As an _offline_ augmentation:
>     - It would require to pre-train a vision-only 3DGS model, splitting the overall training back into two phases and thus slowing it down (see Table 8).
>     - The benefits of offline augmentation are usually limited compared to online augmentation (the former only adds a fixed/limited number of images to the training set, whereas our augmentation scheme provides new randomized language maps at every training iteration). Please refer to the **new Table 15** [Reviewer `Awye`] to see a comparison of our online mix-up strategy with a more realistic offline solution.
>
>
> ----
> ## Q3. Further disentangling visual and language Gaussian representations
>
> We thank the reviewer for another thoughtful question. The main reason why prior 3DGS-based or NeRF-based language-embedded scene representations subordinate the semantic modality to the visual one is that the 2D semantic maps are too sparse to regress the underlying 3D scene geometry from them. The visual/shading information provided by the corresponding color images is required for the 3D representation to properly converge. Moreover, for some downstream tasks such as 3D editing, it is necessary to maintain the same underlying geometry across modalities.
> For these reasons, we decided to keep all spatial properties of the 3D Gaussians (i.e., mean position and covariance) common to the two modalities, and to disentangle only the modality-specific parameters (radiance vs. language vectors, opacity vs. semantic indicator).
>
> We substantiate the above with a new experiment, shared in **Table 19**, Appendix A.3.5. We evaluate multiple variations of our solution over the open-vocabulary semantic segmentation task, with different levels of disentanglement between the represented modalities. As expected, the model with completely separate Gaussians for each modality (similar to training a semantic-only 3DGS model for the segmentation task) fails to effectively learn the underlying 3D information and converges poorly, impacting the downstream results on unseen views. Our solution performs the best out of the four.

---

> ### Author Response · Authors · 2024-11-21
>
> --------------------------
> ----
>
> _(we copy some of the new relevant tables form the Appendix below, to facilitate the reviewer's task.)_
>
> **Table 19.** Ablation results on different levels of disentanglement between the per-modality Gaussian parameters, evaluated on the downstream open-vocabulary semantic-segmentation task on LERF scenes.
>
> |Params. shared across modalities:                  | Position | Covariance | Opacity/Indic. | Features | `ramen` | `figurines` | `teatime` | `kitchen` | avg.  |
> |------------------|----------|------------|----------------|----------|---------|-------------|-----------|-----------|-------|
> |      | ✓ | ✓ | ✓     | ✗ | 55.9    | 71.0        | 54.2      | 51.3      | 58.1  |
> | (ours)           | ✓ | ✓ | ✗         | ✗ | **61.4** | **73.5**   | **58.1**  | **54.8**  | **62.0** |
> |          | ✓     | ✗     | ✗         | ✗ | 47.8    | 65.6    | 42.3    | 40.9    | 49.2  |
> |          | ✗     | ✗     | ✗         | ✗ | 41.4    | 60.7        | 39.2      | 38.8      | 45.0  |

---

> > ### Author Response · Authors · 2024-11-21
> >
> > PS: We have finalized the experiment presented in **Table 19**, and updated our PDF manuscript and the above comments accordingly. We thank the reviewer again for suggesting this interesting evaluation.

---

### Official Review · Reviewer_yCDX · 2024-10-28

**Soundness:** 3
**Presentation:** 3
**Contribution:** 3
**Rating:** 8
**Confidence:** 4

**Summary:**

This paper proposes an interesting idea on how to fuse multi-modality info, i.e. visual & semantic, for better semantic rasterization in 3DGS. This solution mainly composes of a cross-modal rasterizer, a specially designed semantic indicator and a camera-view blending method. The solution is interesting especially because it offers a better solution for semantic rasterization for reflective or translucent objects, which is a serious problem with the original 3DGS. For me, the designed semantic indicator in the Gaussian representation is novel and useful for the claimed main problem. The main contributions also include the empirical quality result of the semantic indicator on these reflective or translucent objects, and quantity comparison of open-vocabulary semantic segmentation tasks. These results conclude the usefulness of the proposed solution.

The paper is generally well written and structured, easy to follow and read. I don’t come across many issues for further clarification. But here a few things I may need the authors to respond. Given those issues properly clarified, I would be willing to increase the score.

1、	Equation (3), any language/semantic should not change due to view angle, so F^W doesn't make sense to me, even the authors mention that superscript W will be omitted for ease of reading. It shouldn’t appear in the original equation at all. Also in the same equation, there lacks a clear definition of what L_sem is, but this should be a minor issue.
2、	Figure 3 right, I’m not 100% sure where the symmetry of the density distribution is from, as there lacks sufficient correlation between l and o.
3、	Table 8, an average training and inference time should be reported on all scenes of a whole dataset, instead of on just one.
4、	The Related Work session is relatively weak and could be fortified with more related reference included. For instance, “HUGS: Holistic Urban 3D Scene Understanding via Gaussian Splatting”, which proposes another way of using semantic info in 3DGS (not traditional 2d way mentioned by the authors）and is worthy of being cited.
5、	A minor grammar error of Line 356-357.

**Strengths:**

The proposed idea is interesting, especially because it offers a better solution for semantic rasterization for reflective or translucent objects, which is a serious problem with the original 3DGS. For me, the designed semantic indicator in the Gaussian representation is novel and useful for the claimed main problem. The main contributions also include the empirical quality result of the semantic indicator on these reflective or translucent objects, and quantity comparison of open-vocabulary semantic segmentation tasks. These results conclude the usefulness of the proposed solution.

**Weaknesses:**

A few issues are listed in the next section.

**Questions:**

1、	Equation (3), any language/semantic should not change due to view angle, so F^W doesn't make sense to me, even the authors mention that superscript W will be omitted for ease of reading. It shouldn’t appear in the original equation at all. Also in the same equation, there lacks a clear definition of what L_sem is, but this should be a minor issue.
2、	Figure 3 right, I’m not 100% sure where the symmetry of the density distribution is from, as there lacks sufficient correlation between l and o.
3、	Table 8, an average training and inference time should be reported on all scenes of a whole dataset, instead of on just one.
4、	The Related Work session is relatively weak and could be fortified with more related reference included. For instance, “HUGS: Holistic Urban 3D Scene Understanding via Gaussian Splatting”, which proposes another way of using semantic info in 3DGS (not traditional 2d way mentioned by the authors）and is worthy of being cited.
5、	A minor grammar error of Line 356-357.

---

> ### Author Response · Authors · 2024-11-21
>
> ## Q1. Notations $F^W$ and $\mathcal{L}_{\text{sem}}$
>
> While the 3D language embeddings $f^i$ (i.e., language vectors attached to each 3D Gaussian $i$) indeed do not change with the view angle, the rasterized 2D semantic features $F^W$ do depend on the view $W$.
> For a given pixel location $v$ (e.g., coordinates (100, 100) in the image), the language embedding $F(v)$ assigned to it will change depending on where the camera is looking (e.g., if the camera is placed in front of cup, then pixel (100, 100) will be assigned a cup-related language vector, but this will change if the camera faces a different object). We apologize for not making it clearer in the paper and we will try to improve.
>
> Regarding $\mathcal{L}_{\text{sem}}$, we agree that an example of a loss function typically used in prior works could be provided. We have edited the manuscript accordingly. Thank you for the suggestion!
>
> ----
> ## Q2. Distribution symmetry in Figure 3
>
> We would like to kindly point out that the distribution of $l^i - o^i$ for the `ramen` scene, shown in Figure 3, is actually _not_ symmetric. E.g., the peak near 1.0 is higher than that near -1.0; the values around -0.10 are higher than those around +0.10; etc. It should also be noted that the overall opacity and semantic-indicator values of the 3D Gaussians depend on the content of each scene.
>
> With the example shared in Figure 3, our goal is simply to highlight that there are scenes/objects that would greatly benefit from disentangling opacity and semantic indicator. I.e., for these scenes/objects, their Gaussian representation is parametrized by significantly different $l^i$ and $o^i$.
>
>
> ----
> ## Q3. Efficiency analysis on all scenes
>
> We provide the results on all the other scenes of the LERF dataset in Appendix A.5.2, c.f. **new Tables 21-24**. These results confirm that our method outperforms prior art in terms of computational and memory efficiency, with an average training time of $73.6$ min per scene and an average FPS of $76.2$.
>
> ----
> ## Q4. HUGS reference
>
> We thank the reviewer for their suggestion, and we have added a reference to HUGS into the Related Work section of our paper.
>
> ----
> ## Q5. Grammar error
>
> We thank the reviewer for their appreciation of our writing and paper structure.
> We have run a grammar check on L356-357, but it did not return any error. Did the reviewer mean another line? Note that we will perform a grammar and spell check of the manuscript once more when preparing the final version.
>
>
> --------------------------
> ----
>
> _(we copy some of the new relevant tables form the Appendix below, to facilitate the reviewer's task.)_
>
> **Tables 21-24.** Efficiency and image-quality analysis on LERF’s data, averaged over all scenes.
> | Method        | Training Time ↓ | FPS ↑ | Gaussians # ↓ | Storage Size ↓ |
> |---------------|-----------------|-------|---------------|----------------|
> | LangSplat     | 104min | 41 | 112k | 302MB |
> | GS-Grouping   | 133min | 74 | 139k | 359MB |
> | GOI           | 83min | 42 | 122k | 329MB |
> | Ours          | **73min** | **76** | **105k** | **296MB** |

---

> > ### Comment · Reviewer_yCDX · 2024-11-26
> >
> > All my previous round of questions have been responded properly by the authors already.  Especially, Tables 21-24 gives an efficiency and image-quality analysis on LERF’s data, which is done by average over all scenes.
> > I recommend this paper should be accepted.

---

> > > ### Author Response · Authors · 2024-11-30
> > >
> > > We are greatly appreciative of the reviewer's continuous support of our submission. As suggested, we will replace the current Table 8 in our paper with the new above table containing results averaged over all scenes, and we shall share all the per-scene Tables in Appendix. Thank you!

---

### Official Review · Reviewer_Jaap · 2024-11-01

**Soundness:** 3
**Presentation:** 3
**Contribution:** 3
**Rating:** 6
**Confidence:** 3

**Summary:**

This paper learns language embedded 3dgs field and addresses the often-neglected issue of appearance-semantic misalignment in rasterization. It introduces a camera-view blending technique to enhance semantic consistency between existing and synthesized views. The experimental findings demonstrate that the proposed approach outperforms existing methodologies.

**Strengths:**

1. The paper provides a profound insight into the challenges posed by semi-opaque media and intricate light transport effects, highlighting the limitations in translating color opacity to the semantic domain. This observation is interesting.
2. The implementation of a single learnable parameter to replace the conventional shared color opacity parameter is both straightforward and efficient.
3. The paper is commendably structured, presenting a well-motivated narrative and a clear methodology.

**Weaknesses:**

1. The paper introduces the camera-view blending technique as a method to enhance cross-view semantic consistency. However, it falls short in fully addressing the challenge of different objects sharing similar colors, potentially leading to indistinguishable semantic representations, which the authors claim to have tackled. Further insights on this critical point are needed.
2. In Section 4.4, the experimental results suggest an increase in rendering speed with the proposed pipeline. However, given the introduction of additional steps and parameters, one would expect a potential decrease in inference speed. The discrepancy in the reported results requires clarification.
3. The title of the paper, "3D Vision-Language Gaussian Splatting," might be misleading as it predominantly focuses on learning open language semantic fields rather than visual appearances. A revision to better reflect the core contribution of the research is recommended.

**Questions:**

I am curious about the impact of semantic learning on rendering quality.

---

> ### Author Response · Authors · 2024-11-21
>
> ## W1. Evaluation of contributions on objects sharing similar colors
>
> We thank the reviewer for their remark. We agree that our submission would benefit from providing more insight w.r.t. the impact of our contributions on specific challenges, such as scenes containing objects with similar appearances but different language embeddings.
>
> Therefore, we added two figures to our manuscript, analyzing the impact of the cross-view regularization brought by our augmentation technique on objects with similar colors/textures, as well as on objects composed of arts with inconsistent appearance:
> - **Figure 18** shows examples of multiple objects sharing the same colors. E.g. in the first scene, the scissors and spatula share the same color, leading part of the scissors to be misclassified as spatula when camera-view blending is not applied. Similarly, in the second scene, the pot and table share the same wooden texture, causing part of the pot to be misidentified as table by the model without camera-view blending.
> - **Figure 19** shows examples of objects composed of multiple colors/textures. In the first example, the camera and the spatula are composed of parts with different colors, and some of these parts end up misclassified by the model without camera-view blending. The same can be observed in the second example: the bike is composed of parts with varied appearances, and the model optimized without our augmentation misses many of them; whereas the model optimized with correctly identifies the whole object.
>
>
> ----
> ## W2. Rendering speed
>
> Among our contributions, only the cross-modality fusion operation and semantic operator are applied at inference time, and both have a negligible computational impact:
> - Our fusion module is simply composed of a self-attention operation and barely takes $\sim$5-7 milliseconds.
> - Our semantic indicator replaces the opacity value passed to the rasterization function during language-embedding rendering. Therefore, it has no computational/dimensional impact on the inference (this additional parameter only impacts the memory footprint of the model).
>
> The faster inference speed compared to other methods comes from the integration of RGB image rasterization and language-feature rasterization into a single stage in our custom CUDA implementation (instead of the sequential RGB-then-language rasterization performed in prior works). This integration explains why our method, along with GS-Grouping (Ye et al., 2024), outperforms other approaches significantly on the FPS metric in Table 8. Note that the reason why our method is faster than GS-Grouping probably lies in the lower number of Gaussians that our model needs to represent the scene, as rendering time scales with that number.
>
> ----
> ## W3. Succinct title
>
> We appreciate the reviewer's suggestion. We agree that our title may be a bit short and overly catchy. As implicitly mentioned in the abstract and introduction, our justification is that, unlike prior works that subordinates the language modality to the appearance one, we are the first to focus on striking a balance between the two modalities; and thus the first to propose an actual "_vision-language Gaussian splatting model_", instead of a "_vision model extended to language_".
> We are brainstorming a more explicit title, maybe along the lines of "_3D Vision-Language Gaussian Splatting: Restoring the Balance between Modalities_" (?).
>
> ----
> ## Q1. Impact of semantic learning on rendering quality
>
> We have added PSNR and SSIM results to **Table 25** in Appendix A.5.2, i.e., for various scenes of the LERF dataset.
> While this work focuses on semantic accuracy, our solution maintains competitive performance w.r.t. image quality, compared to the state-of-the-art, highlighting that our effort to strike a balance between the two modalities did not penalize the visual one. Moreover, comparing to the results from color-only 3DGS (same as LangSplat as this method fixes all 3DGS parameters after its pre-training), we observe that semantic learning does not have an obvious impact on color-modality metrics (i.e., image quality).
>
> We thank the reviewer for giving us the opportunity to present these results and to improve our submission accordingly.

---

> > ### Author Response · Authors · 2024-11-21
> >
> > --------------------------
> > ----
> >
> > _(we copy some of the new relevant tables form the Appendix below, to facilitate the reviewer's task.)_
> >
> >
> > **Table 25.a**. Evaluation of color-image rendering quality on the LERF dataset, in terms of PSNR.
> >
> > | Method             | `ramen` | `teatime` | `figurines` | `kitchen` | avg. |
> > |---------------------|---------|-----------|-------------|-----------|------|
> > | LangSplat (3DGS)   | 28.5    | **32.6**  | **27.9**    | **32.4**  | 30.4 |
> > | GS-Grouping        | **29.2**| 32.1      | 26.5        | 32.3      | 30.0 |
> > | GOI                | 29.0    | 32.3      | 27.7        | 31.9      | 30.2 |
> > | Ours               | **29.2**| 32.5      | **27.9**    | 32.3      | **30.5** |
> >
> > **Table 25.b**. Evaluation of color-image rendering quality on the LERF dataset, in terms of SSIM.
> >
> > | Method             | `ramen` | `teatime` | `figurines` | `kitchen` | avg. |
> > |---------------------|---------|-----------|-------------|-----------|------|
> > | LangSplat (3DGS)   | **0.87**| **0.83**  | 0.76        | 0.85      | **0.83** |
> > | GS-Grouping        | 0.85    | 0.82      | 0.74        | 0.84      | 0.81 |
> > | GOI                | 0.83    | 0.83      | **0.77**    | 0.82      | 0.81 |
> > | Ours               | 0.86    | **0.83**  | 0.76        | **0.86**  | **0.83** |

---

### Official Review · Reviewer_EPr2 · 2024-11-04

**Soundness:** 2
**Presentation:** 3
**Contribution:** 3
**Rating:** 6
**Confidence:** 3

**Summary:**

This paper claims that the performances of current multi-modal scene understanding approaches are limited by the the imbalance between visual and language modalities, which have widely different properties.
To alleviate these limitations, this paper proposes several strategies, including a cross-modal rasterizer that places greater emphasis on language features and a camera-view blending technique.
Finally, the proposed methods achieves state-ofthe-art performance in open-vocabulary semantic segmentation tasks.

**Strengths:**

1. The motivation of this paper  sounds reasonable. The over-fitting on the color modality may have a negative impact on semantic learning, which is consistent with intuition.
2. The proposed techniques, including the smoothed semantic indicator  and the mix-up augmentation, are simple but effective.
3. The performance of the method  outperform existing methods by a significant margin.

**Weaknesses:**

1. Although the motivation  sounds reasonable and the Figure 3 explain the motivation to some extent, I expect more experiments to further study the differences between color and semantics modalities. Beside showing quantitative, qualitative results and ablation of the proposed strategies, the authors should further discuss the deeper mechanisms. For example, the authors can study the relationship between color over-fitting phenomenon and specific scenes.
2. The figure 2 is hard to understand. The method is simple but the process in the figure looks complicated. Some fonts are too small.

**Questions:**

1. What is the specific relationship between the improvement brought by your method and the details of the scene? In what kind of scene is your improvement more obvious?

---

> ### Author Response · Authors · 2024-11-21
>
> ## W1. Differences and relationships between color and semantic representations
>
> We thank the reviewer for their suggestion. Additional experiments and discussions have been added to the Appendix:
>
> - In **new Tables 26-30**, we analyze the impact of disentangling appearance and language rasterization on translucent and reflective objects (in terms of mIoU for the open-vocabulary segmentation task). As discussed with Reviewer `Awye` and in Appendix A.5.3, in conjunction with Figure 3, we observe that translucent and reflective objects tend to be represented with Gaussians having low opacity values. After introducing our semantic indicator to replace the opacity during language rasterization, we note a large difference in the opacity values (low) and semantic-indicator values (high) learned by these Gaussians (e.g., see large $l - o$ results for the translucent glass or reflective bottle/cup in Figure 3). The improvement on downstream semantic tasks, e.g., open-vocabulary semantic segmentation, is thus especially _large for these categories_. E.g., the segmentation of glass is improved by $14.5\%$ and sake cup by $14.3\%$, whereas the segmentation of non-translucent/non-reflective is improved by $3.6\%$ on average, c.f . Table 26.
> - In **new Figures 18-19** in Appendix A.4.4, we qualitatively demonstrate the impact of color over-fitting over specific scenes. As discussed with Reviewer `Jaap`:
>   - Figure 18 shows examples of multiple objects sharing the same colors. E.g. in the first scene, the scissors and spatula share the same color, leading part of the scissors to be misclassified as spatula when camera-view blending is not applied. Similarly, in the second scene, the pot and table share the same wooden texture, causing part of the pot to be misidentified as table by the model without camera-view blending.
>   - Figure 19 shows examples of objects composed of multiple colors/textures. In the first example, the camera and the spatula are composed of parts with different colors, and some of these parts end up misclassified by the model without camera-view blending. The same can be observed in the second example: the bike is composed of parts with varied appearances, and the model optimized without our augmentation misses many of them; whereas the model optimized with correctly identifies the whole object. The proposed technique enhances semantic consistency across diverse views through regularization.
>
>
> ----
> ## W2. Clearer Figure 2
>
> Following the reviewer's remark, we have added **Figure 9** to Appendix A.1 (implementation details), depicting parts of the original Figure 2 with larger fonts and spacing.
>
> ----
> ## Q1. Scene content benefiting from our contributions
>
> We thank the reviewer for their interesting question. As mentioned in our above **W1** response, our introduction of the semantic indicator especially benefits object categories with complex light interactions (e.g., translucent or reflective materials), since prior work suffer from re-using the Gaussians' opacity values to rasterize the language maps (see Figure 3 and Tables 26-30 for further discussions). Furthermore, the cross-view regularization introduced by our view-blending augmentation technique appears to benefit object categories sharing similar colors, as well as object categories composed of multiple parts with different colors/textures, as empirically shown in Figures 18-19 and discussed in Appendix A.4.4.
>
> We would like to add that it would be interesting to measure the performance of existing vision-language Gaussian splatting models as a function of the complexity of the scenes (in terms of increasing number of depicted objects and/or texture complexity), but the literature lacks such a benchmark. Collecting and annotating  3D scenes with increasing complexity to build such a benchmark would greatly benefit the community, we believe (though it goes beyond the scope of our current work).
>
> ----
> ----
> _(we copy some of the new relevant tables form the Appendix below, to facilitate the reviewer's task.)_
>
> **Table 26**. Per-category mIoU results for the open-vocabulary semantic segmentation task on LERF's `ramen` scene. Categories with a "*" are object with translucent/reflective properties.
>
> | Semantic Indicator? | `chopsticks` | `egg`  | `nori` | `napkin` | `noodles` | `kamaboko` | `onion_segments` | `corn` | `bowl`* | `sake_cup`* | `plate`* | `glass`* | `spoon`* |
> |---------------------|--------------|--------|--------|----------|-----------|------------|------------------|--------|---------|------------|----------|----------|----------|
> | ✗  | 90.0 | 96.4 | 78.2 | 45.9 | 51.4  | 37.7 | 24.7 | 10.4 | 40.3  | 36.7 | 53.6 | 64.1 | 30.3 |
> | ✓  | 90.4 | 96.5 | 80.4 | 51.2 | 56.6  | 44.5 | 29.8 | 13.9 | 53.7  | 51.0 | 62.1 | 78.6 | 45.4 |
> | Improvement  | +0.4 | +0.1 | +2.2 | +5.3 | +5.2  | +6.8 | +5.1 | +3.5 | +13.4 | +14.3  | +8.5 | +14.5  | +15.1  |
>
> (Tables 27-30 in the Appendix provide similar results on other LERF scenes.)

---

> > ### Comment · Reviewer_EPr2 · 2024-11-26
> >
> > The authors' response has effectively addressed my concerns.
> > The additional experiments demonstrate that the proposed method achieves greater improvements, particularly on objects with translucent or reflective properties. This result can validate the motivation of the paper very well.
> >  I recommend accepting this paper.

---

> > > ### Author Response · Authors · 2024-11-30
> > >
> > > We deeply thank the reviewer for acknowledging our responses and recognizing that "_the proposed method achieves greater improvements, particularly on objects with translucent or reflective properties._"
> > >
> > > We are grateful for the reviewer's suggestions (c.f. new Tables 26-30, Figures 18-19) that have strengthened our submission, and for the reviewer's acceptance recommendation.

---

### Official Review · Reviewer_Awye · 2024-11-04

**Soundness:** 3
**Presentation:** 2
**Contribution:** 2
**Rating:** 6
**Confidence:** 2

**Summary:**

This paper focuses on improving the semantic rendering quality within well-reconstructed 3D scenes. With introducing self-attention, semantic indicator and regularization, the  proposed method achieves better semantic segmentation results, especially for
transparent objects.

**Strengths:**

1. The paper is well-written and easy to follow.

2. The designs are well-motivated.

3. The performance shows a significant improvement.

**Weaknesses:**

(1) Line 215, the u^i is the fused features or position?

(2) In camera interpolation, why use mix-up to generate a feature map instead of using an off-the-shelf OV model to output one? Just like previous methods.

(3) Missing a global ablation study to present the importance of 'self-attention', 'semantic indicator'. 'camera interpolation' over your baseline. Now, it is unclear how the baseline stands in comparison with previous methods and the contributions from the three proposed designs.

(4)  Low efficiency, the Table. 8 shows it takes over one hour to learn the semantics of single 3D scenes, which is significantly longer than typical 3D Gaussian reconstruction (which may take less than 10 mins on A100？), especially considering the iteration is only 15k. What makes the rasterization so slow? In addition, since more attributions are added in Gaussian, it is fair to provide the storage cost of Gaussian models when compared with other works in Table.8 instead of only Gaussian numbers.

(5) The incomplete experiments. The experiments on 3D-OVS are conducted on  5/7 scenes instead of full scenes.

(6) No deep analysis of performance. In my view, the IoU improvements are mainly from the translucent or reflective objects, as shown in Figure.3, no detailed statistics to prove that.

**Questions:**

(1) What is the different between self-attention and cross-attention mechanism?

(2) Considering the experiment setting is the open-vocabulary, for example, the CLIP feature dimension is 768, how to obtain 768 rendering with d_c+d_f=6? How to perform the dim. Reduction as shown in Figure.2

---

> ### Author Response · Authors · 2024-11-21
>
> ## W1. Role of $u^i$
>
> The notation $u^i$ represents the fused features, i.e., the concatenated color features and language features passed to the rasterizer.
>
> ----
> ## W2. Mix-up vs. 3DGS+CLIP-based data augmentation
>
> We agree with the reviewer that one could apply the selected off-the-shelf language-feature extractor (e.g., CLIP) to generate new pseudo-ground-truth features. This could be done either:
> - _online_, i.e., using the model's predicted color image as input to the language-feature extractor in order to generate the target language map to evaluate the one rendered by the model.
> - _offline_, i.e., using a pre-trained color-only 3DGS model of the scene to generate novel views, pass these views to the feature extractor, and use all the resulting data to train the language-embedded representation.
>
> However, the computational cost of running the selected language-feature extractor (CLIP) is too heavy for online usage, and the offline strategy also comes with several downsides.
> First, one of our contributions is our joint CUDA-based rasterization function, which enables the fast rendering of both modalities (color and language). While most prior works (Qin et al., 2024) train their model in two phases (a first phase to train a color-only 3DGS model, then a second phase to extend it with language information), we leverage our custom rasterizer to efficiently optimize our vision-language model in a single training phase (see Table 8 highlighting our training and rendering efficiency). The downside of this faster training is that, unlike prior solutions, we do not have access to a pre-trained 3DGS model of the target scene. Adding a pre-training phase for the color-only model would have a _significant computational overhead_.
> Second, the aforementioned strategy (i.e., rendering additional views using a pre-trained 3DGS) only adds a fixed/limited number of images to the training set (c.f. _offline_ rendering before the training starts), whereas our augmentation scheme provides new randomized language maps at every training iteration (c.f. _online_ augmentation).
>
> We quantitatively compared the suggested off-the-shelf augmentation strategy to ours in a new experiment, which confirms that our augmentation technique does indeed result in a more robust representation w.r.t. downstream semantic tasks. Please refer to the **new Table 15** and corresponding discussion added to Appendix A.3.4. Note that in future work, it would be interesting to investigate how both strategies could be optimally interleaved (e.g., by pretraining our model with mix-up augmentation;  then, after some iterations, augmenting the training data with 3DGS/CLIP-rendered pairs).
>
>
> ----
> ## W3. Global ablation study
>
> We thank the reviewer for the suggestion. We have added a global ablation study on two datasets (LERF and 3D-OVS) presented in **Tables 11-12** of Appendix A.3.1. The results illustrate that each of the three components plays a crucial role in improving the performance.
>
>
> ----
> ## W4. Training efficiency of vision-language 3DGS models
>
> The provided time values are high since they include pre-training phases, such as the extraction of language features (i.e., applying CLIP to input images) and the training and inference of the auto-encoder to compress the said CLIP features.
>
> It should be highlighted that the training times shown in Table 8 are actually in the _expected range_ for Gaussian splatting model covering both RGB and language modalities (i.e., much slower than RGB-only models due to higher dimensionality, dual rasterization, etc.). Related works share similar training times as the ones that we measured — see Table 3 in the GOI paper (Qu et al., 2024) for example.
>
>
> ----
> ## W5. 3D-OVS scene subset
>
> We follow the protocol adopted in LangSplat (Qin et al., 2024), where experiments are performed on the five most relevant scenes. While the 3D-OVS indeed contains a total of ten scenes, even the dataset authors only use six scenes in their paper presenting 3D-OVS. While we believe that it is thus reasonable to evaluate on 5 scenes, we agree with the reviewer that the more results, the better.
>
> We thus provide _new results_ on two extra scenes, `table` and `snacks` in **Table 20** of Appendix A.5.1. The results shared there confirm the effectiveness of our solution in terms of open-vocabulary semantic segmentation.

---

> ### Author Response · Authors · 2024-11-21
>
> ----
> ## W6. Detailed statistics for translucent and reflective objects
>
> We thank again the reviewer for their constructive suggestion. We agree that our paper could benefit from showing results for each object category separately, to highlight how objects with translucent/reflective properties benefit the most from our semantic-indicator contribution. We have performed this evaluation on various scenes, and we have added the corresponding results and discussion to Appendix A.5.3 (see **Tables 26-30**).
>
> As shown in Figure 3, translucent and reflective objects are mostly represented by
> 3D Gaussians with low opacity values (in order to model their complex light transport). Using these low opacity values to rasterize the language features cause unwanted artifacts. This is alleviated by our semantic indicator. Our model correctly learns higher indicator values for Gaussians representing the above-mentioned object categories (e.g., see large $l - o$ results for the translucent glass or reflective bottle/cup in Figure 3). The improvement on downstream semantic tasks, e.g., open-vocabulary semantic segmentation, is thus _especially large for these categories_. E.g., the segmentation of glass is improved by $14.5\%$ and sake cup by $14.3\%$, whereas the segmentation of non-translucent/non-reflective is improved by $3.6\%$ on average, c.f . Table 26.
>
>
> ----
> ## Q1. Self-attention vs. cross-attention
>
> Our framework leverages cross-modality self-attention, where query, key, and value are all $c^i \oplus f^i$ as shown in Equation 5.
> In the ablation on cross-modal rasterizer (Section 4.3), the cross-attention fusion tested in Table 4 is performed by using $f^i$ as query and $c^i$ as both key and value.
>
>
> ----
> ## Q2. Dimension reduction
>
> For the extraction and compression of the language features, we follow LangSplat's pipeline (Qin et al., 2024), c.f. L157-161. I.e., an auto-encoder is trained to compress the scene-relevant CLIP features from their original size (768) to a much more compact $d_f$ target (with $d_f=3$ in LangSplat and our work).
>
> Afterwards, both the Gaussian splatting model and the downstream tasks only rely on the compressed feature space. E.g., for open-vocabulary downstream tasks, the input queries are first encoded by the same auto-encoder network, before being compared to language features generated by the Gaussian splatting model.
>
> We hope that this answers your question. We will further clarify in the manuscript.
>
> ----
> ----
>
> _(we copy some of the new relevant tables form the Appendix below, to facilitate the reviewer's task.)_
>
> **Table 15.** Comparison between data-augmentation strategies, over downstream open-vocabulary semantic segmentation (mIoU) on LERF scenes. Note that the original training sets contain between 124 and 297 images per scene. For the offline augmentation using 3DGS-rendered images and off-the-shelf open-vocabulary models (SAM/CLIP), 120 novel views are added, with their viewpoints sampled via interpolation.
>
> | Method  | `ramen`  | `figurines` | `teatime` | `kitchen` | avg.  |
> |--|--|-|-|-|-|
> | w/o data augmentation  | 53.3 | 66.9  | 46.3  | 45.3  | 53.0  |
> | w/ data from 3DGS & SAM/CLIP | 55.2 | 68.1  | 49.5  | 48.2  | 55.3  |
> | w/ view blending  (ours) | **61.4** | **73.5** | **58.1** | **54.8** | **62.0** |
>
> **Table 20.** Evaluation on 3D-OVS dataset extended to additional scenes.
>
> | Method  | Venue | `bed`  | `bench`  | `room` | `sofa` | `lawn` | `table`  | `snacks` | avg. |
> |--|-|--|--|--|--|--|--|--|--|
> | Feature-3DGS  | CVPR'24  | 83.5 | 90.7 | 84.7 | 86.9 | 93.4 | 92.4 | 88.6 | 88.6 |
> | LEGaussians | CVPR'24  | 84.9 | 91.1 | 86.0 | 87.8 | 92.5 | 93.6 | 87.5 | 89.1 |
> | LangSplat | CVPR'24  | *92.5* | *94.2* | *94.1* | *90.0* | *96.1* | 95.9 | *92.3* | *93.6* |
> | GS-Grouping | ECCV'24  | 83.0 | 91.5 | 85.9 | 87.3 | 90.6 | 94.2 | 88.7 | 88.8 |
> | GOI | ACMMM'24 | 89.4 | 92.8 | 91.3 | 85.6 | 94.1 | 95.6 | 89.3 | 91.1 |
> | FMGS  | IJCV'24  | 80.6 | 84.5 | 87.9 | 90.8 | 92.6 | *97.2* | 83.0 | 88.2 |
> | Ours  |  | **96.8** | **97.3** | **97.7** | **95.5** | **97.9** | **98.5** | **94.7** | **97.0** |
>
> **Table 26**. Per-category mIoU results for the open-vocabulary semantic segmentation task on LERF's `ramen` scene. Categories with a "*" are object with translucent/reflective properties.
>
> | Semantic Indicator? | `chopsticks` | `egg`  | `nori` | `napkin` | `noodles` | `kamaboko` | `onion_segments` | `corn` | `bowl`* | `sake_cup`* | `plate`* | `glass`* | `spoon`* |
> |-|--|--|--|--|-|--|--|--|-|--|--|--|--|
> | ✗  | 90.0 | 96.4 | 78.2 | 45.9 | 51.4  | 37.7 | 24.7 | 10.4 | 40.3  | 36.7 | 53.6 | 64.1 | 30.3 |
> | ✓  | 90.4 | 96.5 | 80.4 | 51.2 | 56.6  | 44.5 | 29.8 | 13.9 | 53.7  | 51.0 | 62.1 | 78.6 | 45.4 |
> | Improvement  | +0.4 | +0.1 | +2.2 | +5.3 | +5.2  | +6.8 | +5.1 | +3.5 | +13.4 | +14.3  | +8.5 | +14.5  | +15.1  |
>
> (Tables 27-30 in the Appendix provide similar results on other LERF scenes.)

---

> > ### Comment · Reviewer_Awye · 2024-11-26
> >
> > Thanks for the authors' responses, which well address my concerns.
> >
> > I strongly suggest supplementing these analyses in the revision.

---

> > > ### Author Response · Authors · 2024-11-30
> > >
> > > We are grateful for the reviewer's constructive comments and the time invested in evaluating our work. We sincerely appreciate the thoughtful feedback and kind acknowledgment of our efforts to address the questions raised during the review process. We will make sure to account for these remarks in the revision. Thank you!

---

### Author Response · Authors · 2024-11-21
**Global Response to Reviewers**

We are very grateful to the reviewers `Awye`, `EPr2`, `Jaap`, `yCDX`, and `56RB` for the constructive feedback, as well as the recognition of our paper's strengths, such as its novelty and effectiveness.

In this global response, we further highlight the strengths of our work shared by reviewers and address the marginal concern and question brought by more than one reviewer. Other individual questions are answered in our per-reviewer responses, and we are looking forward to further discussing with reviewers.

----
## Shared Strengths

### 1. Novel and Effective Method

Reviewers commended the novelty of our method [`yCDX`, `56RB`], highlighting its intuitive yet highly effective contributions [`EPr2`, `Jaap`, `yCDX`]

### 2. Significant Improvements

All reviewers [`Awye`, `EPr2`, `Jaap`, `yCDX`, `56RB`] acknowledge that our method outperforms existing solutions, with many deeming the improvement as "_significant/promising_" [`Awye`, `EPr2`, `yCDX`, `56RB`].


### 3. Well-motivated and Insightful Work

Reviewers overall agree that our work is well-motivated [`Awye`, `EPr2`, `Jaap`] and tackles a "_serious/overlooked problem_" [`yCDX`, `56RB`] impacting prior works. Reviewer `Jaap` adds that our insight about semi-opaque media and intricate light transport effects is "_profound_".

### 4. Well-written and Structured Paper

Several reviewers [`Awye`, `Jaap`, `yCDX`] praised our manuscript as being structured, well-written, and easy to follow.

----
## Shared Questions

Overall, the questions and concerns brought by the reviewers have little overlap, except for the two listed below. We partly tackle them below and further answer each reviewer in their respective response.

### 1. More Throrough Study of Contributions Impact on Specific Content

Reviewers `EPr2` and `Jaap` fairly remark that we could have more thoroughly studied the impact of our contributions on specific scenes content, such as the positive impact of disentangling color/language rasterization on specific objects with challenging appearances [`EPr2`] or the impact of the cross-view regularization brought by our augmentation technique on objects sharing similar colors [`Jaap`].

We agree that our manuscript could benefit from additional experiments and visualizations highlighting our performance w.r.t. these specific, challenging cases. Therefore, the following experiments and discussions have been added to the Appendix:

- In **new Tables 26-30** and Appendix A.5.3, we analyze the significant impact of disentangling appearance and language rasterization on translucent and reflective objects (w.r.t. downstream semantic tasks). See discussion with Reviewer `Awye` for more details.
- In **new Figures 18-19** and Appendix A.4.4, we qualitatively highlight and discuss the impact of color over-fitting over specific scenes. We show examples of scenes with multiple objects sharing the same colors, and scenes containing objects composed of multiple colors/textures. Our solution with view-blending augmentation clearly outperforms methods without on these objects, as further discussed with Reviewer `Jaap`.


### 2. Question about Online View-blending Augmentation vs. Offline 3DGS+CLIP-based Augmentation

Both Reviewers `Awye` and `56RB` raised the question of using a pre-trained color-only 3DGS model and or the off-the-shelf feature-extractor model to assist with data augmentation, instead of our proposed view-blending technique. In our responses to both, we highlight how our online augmentation scheme is not only easier to adopt to due its light computational overhead, bu that it also results in better cross-view regularization and more robust language representations. We substantiate this claim with a new experiment, added into Appendix A.3.4 (see **new Table 15** and related discussion there).

----
## New Results and Manuscript Edits

We would also like to highlight that we have submitted an updated version of our manuscript, accounting for the reviewers' suggestions and containing additional results (further discussed in our responses to each reviewer).

To facilitate the reviewers' task, we have highlighted all our changes to the manuscript by using a $\color{blue}{\text{\textbf{blue}}}$ font color.

---

### Meta-Review · Area_Chair_VHPT · 2024-12-22

**Metareview:**

This paper presents a 3D vision-language Gaussian splatting model for scene understanding. The proposed method includes a semantic indicator, a cross-modal rasterizer, and a camera-view blending augmentation technique. The paper aims to address limitations in existing methods, particularly challenges with translucent and reflective objects, as well as semantic misalignment due to color overfitting.

The reviewers acknowledge the paper's contributions, rigorous experiments, and potential impacts. There are some initial concerns regarding clarify, experiment details, and method novelty, which are all addressed through additional experiments and thorough response. Therefore, AC recommends accepting this paper.

**Additional Comments On Reviewer Discussion:**

Reviewers raised initial questions on paper novelty, several design choices, and clarify, with no critical concerns. These have been addressed in authors' responses, including ablation studies of each module, results on additional datasets, extra comparisons.

---

### Decision · Program_Chairs · 2025-01-22

Accept (Poster)